# Many Eyes, One Mind: Temporal Multi-Perspective and Progressive Distillation for Spiking Neural Networks

**Kai Sun**[1], **Peibo Duan**[1*] , **Yongsheng Huang**[2], **Nanxu Gong**[3], **Levin Kuhlmann**[1*]
[1] Department of Data Science and AI, Monash University, Melbourne, Australia
[2] School of Software, Northeastern University, Shenyang, China
[3] School of Computing and Augmented Intelligence, Arizona State University, Arizona, USA
{peibo.duan, levin.kuhlmann}@monash.edu

## Abstract

Spiking Neural Networks (SNNs), inspired by biological neurons, are attractive for their event-driven energy efficiency but still fall short of Artificial Neural Networks (ANNs) in accuracy. Knowledge distillation (KD) has emerged as a promising approach to bridge this gap by transferring knowledge from ANNs to SNNs. Temporal-wise distillation (TWD) leverages the temporal dynamics of SNNs by providing supervision across timesteps, but it applies a constant teacher output to all timesteps, mismatching the inherently evolving temporal process of SNNs. Moreover, while TWD improves per-timestep accuracy, truncated inference still suffers from full-length temporal information loss due to the progressive accumulation process. We propose **MEOM** (**M**any **E**yes, **O**ne **M**ind), a unified KD framework that enriches supervision with diverse temporal perspectives via mask-weighted teacher features and progressively aligns truncated predictions with the full-length prediction, thereby enabling reliable inference at arbitrary timesteps. Extensive experiments and theoretical analyses demonstrate that MEOM achieves state-of-the-art performance on multiple benchmarks. Code is available at https://github.com/KaiSUN1/MEOM.

## 1 Introduction

Spiking Neural Networks (SNNs), often regarded as the third generation of neural models (Maass, 1997), differ from traditional Artificial Neural Networks (ANNs) by transmitting information through discrete, sparse, and temporal spikes. The asynchronous, event-driven nature of SNNs enables efficient spatio-temporal pattern encoding with low energy consumption, making them well-suited for deployment on resource-limited platforms such as edge devices and neuromorphic hardware (Indiveri & Liu, 2015; Davies et al., 2018). Operating over discrete timesteps, an SNN incrementally processes input spikes and integrates them temporally into a decision.

Despite these advantages, SNNs typically face a performance gap compared with ANNs, which achieve high accuracy through precise floating-point operations (Roy et al., 2019). To narrow this gap, two main strategies have been explored. The first is ANN-to-SNN conversion, which transfers accuracy effectively but often requires a large number of inference steps, limiting practicality under latency or energy constraints (Rueckauer et al., 2017; Bu et al., 2023). The second is direct training with knowledge distillation (KD), where methods such as KDSNN (Xu et al., 2023) and LaSNN (Hong et al., 2023) distill features or logits from a teacher ANN, enabling competitive results with fewer timesteps. However, these approaches largely overlook the intrinsic temporal dynamics of SNNs. To address this, temporal-wise distillation (TWD) (Yu et al., 2025a) was proposed, which treats outputs across timesteps as a temporal ensemble and supervises each timestep directly. By improving the accuracy of individual timesteps, TWD enhances both overall accuracy and truncated inference performance, which is required under strict latency and energy constraints but inevitably entails accuracy degradation (Li et al., 2023b).

---

*Corresponding author

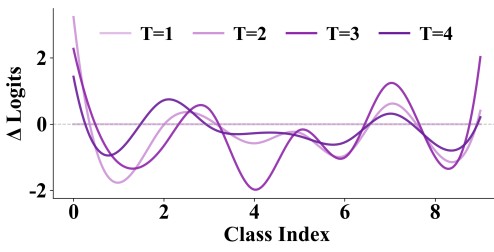 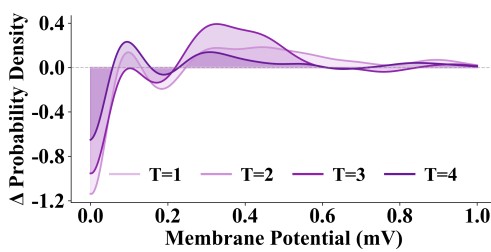

(a) Differential logits relative to T = 1.  (b) Differential voltage distribution relative to T = 1.

Figure 1: Visualization of the temporal difference in SNNs.

Nevertheless, in TWD the teacher ANN's outputs serve as a constant supervisory signal across all timesteps, while the SNN itself evolves dynamically. In practice, outputs across timesteps do not remain identical, as discussed in Appendix A. The final logits themselves also exhibit temporal variation, as illustrated in Figure 1a. Therefore, relying on a fixed ANN target to supervise every timestep may not fully capture the inherent temporal dynamics of SNNs. A more effective strategy would incorporate diverse temporal supervisory signals, aligning more naturally with the time-varying behavior of SNNs, analogous to "many eyes", focusing on different temporal perspectives.

Meanwhile, truncated inference introduces another challenge. As shown in Figure 1b, membrane potentials at $t = 1$ are concentrated at lower values, and they progressively accumulate at later timesteps. As a result, truncated inference can only exploit these early and incomplete signals, inevitably leading to accuracy degradation. Although TWD improves accuracy at individual timesteps, truncated predictions still suffer from temporal information loss. To mitigate this issue, it is important to amplify the density of information within the truncated timesteps during training, allowing the predictions to approximate those obtained from full-timestep inference. Regardless of the extent of truncation, the intended outcome is that the model produces identical predictions, analogous to "one mind", ensuring consistency across both truncated and full-length inference scenarios.

The absence of "many eyes" and "one mind" in KD-based SNN training arise from insufficient learning along the temporal dimension. To address these limitations, we introduce **Temporal Multi-Perspective Distillation (TMPD)**, generating diverse teacher outputs from a single ANN to provide heterogeneous supervision that captures diverse temporal perspectives, and the complementary **Temporal Progressive Distillation (TPD)**, enforcing predictions at each timestep to progressively align with the full-length prediction, ensuring temporal flexibility under truncated inference. Together, TMPD and TPD form our unified framework, **MEOM** (**M**any **E**yes, **O**ne **M**ind). The main contributions of this work are threefold.

- We introduce MEOM, a unified KD framework that integrates TMPD to provide diverse temporal supervision and TPD to align truncated predictions with the full-length prediction, enabling reliable inference across all timesteps.

- We provide theoretical analyses showing that MEOM achieves stable convergence across all timesteps, leading to higher accuracy and better generalization.

- We validate MEOM through comprehensive experiments on CIFAR-10, CIFAR-100, and ImageNet, achieving state-of-the-art performance, together with ablation studies and analyses of energy efficiency and visualization.

## 2 RELATED WORK

### 2.1 KNOWLEDGE DISTILLATION FOR SNNS

Knowledge distillation (KD), which transfers supervision from a high-performing teacher to a student, has become a widely adopted strategy for training SNNs without resorting to ANN-to-SNN conversion. Early **ANN-to-SNN distillation** studies leveraged ANN teachers to provide intermediate feature guidance and/or logit targets (Xu et al., 2023; Hong et al., 2023; Qiu et al., 2024;

Hong & Wang, 2025; Guo et al., 2023; Xu et al., 2024b), achieving competitive accuracy with fewer timesteps compared to conversion-based approaches. Building on this, **SNN-to-SNN distillation** emerged, such as using a longer-timestep SNN to guide a shorter one via temporal-spatial self-distillation (TSSD) (Zuo et al., 2024), or transferring compact yet informative spike representations from a pruned model in Sparse-KD (Xu et al., 2024a), enabling latency reduction without sacrificing representational quality. In parallel, **TWD** treats the spiking sequence as a collection of temporal sub-networks and applies temporal-wise supervision from a single ANN teacher, improving both per-timestep accuracy and the final output (Yu et al., 2025b;a; Konstantaropoulos et al., 2025). Despite these advances, it supervises all timesteps with the same teacher signal, overlooking the intrinsic temporal diversity of spiking dynamics and tending to homogenize intermediate predictions. By contrast, in the ANN, multi-teacher ensemble distillation has been shown to improve both generalization and robustness by aggregating complementary inductive biases from teachers with different architectures or parameterizations (Zhang et al., 2018; Wen et al., 2024; Yang et al., 2025). Motivated by this principle, we design a distillation strategy where each temporal sub-network is guided by distinct teacher features reweighted by masks, restoring diverse temporal perspectives ("many eyes") while achieving the effect of multi-teacher ensembles without training multiple teachers."

## 2.2 TIME FLEXIBILITY FOR SNNS

SNNs accumulate information over multiple timesteps, but real-world neuromorphic deployment often demands low-latency and energy-efficient inference, making it essential to preserve accuracy when the number of available timesteps is reduced. This has motivated methods that improve temporal flexibility from complementary perspectives. **Adaptive truncation** approaches, such as SEENN (Li et al., 2023c) that terminates inference based on confidence (SEENN-I) or a learned policy (SEENN-II), and Pareto-front-based thresholding (Li et al., 2023a), dynamically decide when to stop. **Training-based strategies** aim to strengthen early-timestep predictions without dynamic control, including HSD (Zhong et al., 2024) with task-specific fine-tuning, MTT (Du et al., 2025) that treats timesteps as random variables for augmentation, and SSNN (Ding et al., 2024) that progressively reduces the number of timesteps to lower latency. **Consistency-oriented methods** further encourage stable predictions across timesteps, such as viewing each timestep as an independent sub-model for ensembling (Yu et al., 2025a), enforcing all-pair consistency (Zhao et al., 2025), or aligning temporally adjacent sub-networks (Ding et al., 2025). However, these approaches still rely on pairwise or stepwise consistency, which does not guarantee that early truncated predictions approximate the full-length prediction. As a result, when inference is stopped early, the model may produce immature or inconsistent results. To address this issue, we propose a strategy that progressively aligns truncated outputs with the full-length prediction ("one mind"), ensuring that even under truncated inference, the outputs converge toward the same prediction.

## 3 METHODOLOGY

### 3.1 PRELIMINARIES

Two strategies are commonly used for ANN-to-SNN distillation: Temporal-Averaged Distillation (TAD), aligning with the average prediction, and Temporal-Wise Distillation (TWD), aligning with each timestep output, as shown in Figure 2a and 2b.

**Temporal-Averaged Distillation (TAD).** Let $C$ denote the number of classes and $T$ the number of discrete timesteps. At each step $t \in \{1, \dots, T\}$, the student SNN outputs a pre-softmax logit vector $\boldsymbol{z}_t^S \in \mathbb{R}^C$. These logits are averaged across time to produce a single representative vector $\boldsymbol{z}^{S_{\mathrm{avg}}} = \frac{1}{T} \sum_{t=1}^T \boldsymbol{z}_t^S$. The student probability distribution is given by the temperature-scaled softmax with $\tau > 0$ as the temperature parameter: $\boldsymbol{p}^{S_{\mathrm{avg}}} = \mathrm{softmax}_\tau(\boldsymbol{z}^{S_{\mathrm{avg}}}) = \frac{\exp(z_i^{S_{\mathrm{avg}}}/\tau)}{\sum_{j=1}^C \exp(z_j^{S_{\mathrm{avg}}}/\tau)}$. Supervision on the aggregated output includes hard-label and soft-label components. The hard-label loss uses cross-entropy with ground-truth $\boldsymbol{y} \in \{0,1\}^C$:

$$\mathcal{L}_{\mathrm{TAD\text{-}CE}} = \mathrm{CE}(\boldsymbol{p}^{S_{\mathrm{avg}}}, \boldsymbol{y}) = -\sum_{i=1}^C y_i \log p_i^{S_{\mathrm{avg}}}. \tag{1}$$

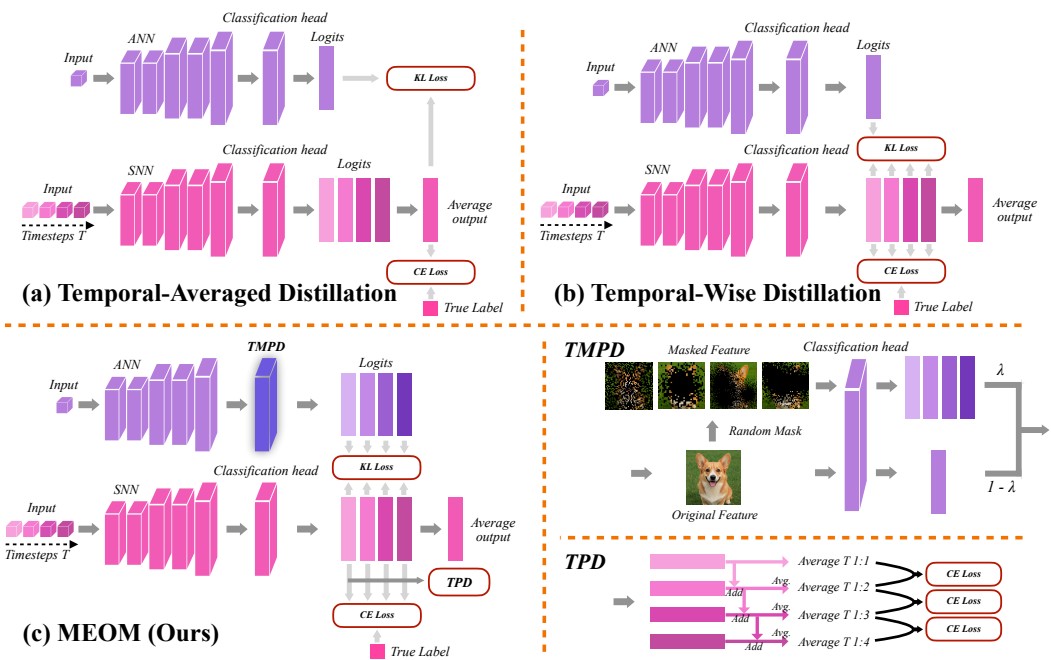

Figure 2: Comparison of temporal knowledge distillation approaches: (a) Temporal-Averaged Distillation, (b) Temporal-Wise Distillation, and (c) Our Proposed MEOM with Related Components.

Soft-label supervision uses the Kullback–Leibler (KL) divergence to align the student's distribution with that of the teacher. Let $\boldsymbol{p}^A$ denote the teacher ANN's probability distribution. The KL loss is

$$\mathcal{L}_{\text{TAD-KL}} = \text{KL}(\boldsymbol{p}^A \| \boldsymbol{p}^{S_{\text{avg}}}) = \sum_{i=1}^{C} p_i^A \log \frac{p_i^A}{p_i^{S_{\text{avg}}}} = -\sum_{i=1}^{C} p_i^A \log p_i^{S_{\text{avg}}} + H(\boldsymbol{p}^A), \quad (2)$$

where the entropy term $H(\boldsymbol{p}^A) = -\sum_{i=1}^{C} p_i^A \log p_i^A$ is constant with respect to the student parameters and is omitted during training. The total TAD loss combines both terms,

$$\mathcal{L}_{\text{TAD}} = \alpha \cdot \mathcal{L}_{\text{TAD-CE}} + \beta \cdot \mathcal{L}_{\text{TAD-KL}}, \quad (3)$$

where $\alpha$ and $\beta$ are hyperparameters controlling the trade-off between hard-label and soft-label.

**Temporal-Wise Distillation (TWD).** TWD leverages the temporal nature of SNN outputs by supervising the student network at each timestep rather than collapsing temporal information into an aggregated vector as in TAD. Let $\boldsymbol{p}_t^S$ denote the probability distribution of the student at timestep $t$. By supervising each timestep directly, this fine-grained strategy enhances per-timestep accuracy, resulting in more stable outputs and a more reliable final prediction. Formally, both hard-label and soft-label losses are applied at every timestep and then averaged over all $T$ steps:

$$\mathcal{L}_{\text{TWD-CE}} = \frac{1}{T} \sum_{t=1}^{T} \text{CE}(\boldsymbol{p}_t^S, \boldsymbol{y}) = \frac{1}{T} \sum_{t=1}^{T} \left( -\sum_{i=1}^{C} y_i \log p_{t,i}^S \right), \quad (4)$$

$$\mathcal{L}_{\text{TWD-KL}} = \frac{1}{T} \sum_{t=1}^{T} \text{KL}(\boldsymbol{p}^A \| \boldsymbol{p}_t^S) = \frac{1}{T} \sum_{t=1}^{T} \left( -\sum_{i=1}^{C} p_i^A \log p_{t,i}^S \right). \quad (5)$$

The final loss combines the two objectives as

$$\mathcal{L}_{\text{TWD}} = \alpha \cdot \mathcal{L}_{\text{TWD-CE}} + \beta \cdot \mathcal{L}_{\text{TWD-KL}}. \quad (6)$$

### 3.2 Temporal Multi-Perspective and Progressive Distillation

While TWD applies identical teacher targets to all timesteps and optimizes them in isolation, it neglects both temporal diversity and truncated information loss. To address this, we introduce MEOM, which integrates TMPD for diverse temporal-wise supervision and TPD for progressive alignment, as shown in Figure 2c.

**Temporal Multi-Perspective Distillation (TMPD).** SNNs produce a sequence of outputs over time, reflecting their intrinsic temporal dynamics. Conventional KD, however, reuses a single static teacher output to supervise all timesteps, which cannot adequately capture the temporal variation in SNN predictions. To overcome this limitation without relying on multiple pre-trained teachers, TMPD draws inspiration from multi-teacher distillation and introduces lightweight mask-based transformations at different timesteps. These transformations diversify the teacher signals while preserving semantic consistency, thereby providing richer and temporally aligned supervision for the student across timesteps.

For each timestep $t \in \{1, \ldots, T\}$, we add a lightweight time-indexed mask $\boldsymbol{m}_t$ ensuring semantic consistency to form a masked feature $\tilde{\boldsymbol{f}}_t^A = \boldsymbol{f}^A \odot \boldsymbol{m}_t$, where $\odot$ denotes the Hadamard product. Passing both the original and masked features through the same classifier yields the original logits $\boldsymbol{z}^A$ and the masked logits $\tilde{\boldsymbol{z}}_t^A$. This process can be written as:

$$\tilde{\boldsymbol{z}}_t^A = \boldsymbol{W}_c\big(\boldsymbol{f}^A + \tilde{\boldsymbol{f}}_t^A\big) = \boldsymbol{W}_c \boldsymbol{f}^A + \boldsymbol{W}_c(\boldsymbol{f}^A \odot \boldsymbol{m}_t) =: \boldsymbol{z}^A + \delta \boldsymbol{z}_t^A, \tag{7}$$

where $\boldsymbol{W}_c$ is the classification weight matrix and $\delta \boldsymbol{z}_t^A = \boldsymbol{W}_c(\boldsymbol{f}^A \odot \boldsymbol{m}_t)$ denotes the perturbation to the teacher logits at timestep $t$. Finally, the original and masked logits are linearly combined to produce temporal-wise perturbed teacher logits:

$$\hat{\boldsymbol{z}}_t^A = (1 - \lambda)\boldsymbol{z}^A + \lambda \tilde{\boldsymbol{z}}_t^A. \tag{8}$$

A temperature-scaled softmax is applied to $\hat{\boldsymbol{z}}_t^A$ to obtain the teacher distribution $\boldsymbol{p}_t^A$, which is compared with the student distribution $\boldsymbol{p}_t^S$ at each timestep using KL divergence averaged over $T$ steps:

$$\mathcal{L}_{\text{TMPD-KL}} = \frac{1}{T} \sum_{t=1}^{T} \text{KL}(\boldsymbol{p}_t^A \parallel \boldsymbol{p}_t^S) = \frac{1}{T} \sum_{t=1}^{T} \sum_{i=1}^{C} p_{t,i}^A \log \frac{p_{t,i}^A}{p_{t,i}^S}. \tag{9}$$

In addition to this KL term, TMPD also adopts the temporal-wise cross-entropy loss $\mathcal{L}_{\text{TWD-CE}}$ (Eq. 4). By introducing mask weighting, TMPD delivers a form of multi-perspective supervision ("many eyes"), where each timestep receives a slightly perturbed but semantically consistent teacher signal, in contrast to TWD's identical targets across steps.

**Temporal Progressive Distillation (TPD).** In neuromorphic deployment, SNNs are often constrained to truncated timesteps due to strict latency and energy requirements, forcing predictions to be made with incomplete temporal evidence. Under such conditions, TAD averages outputs and dilutes discriminative signals, while TWD improves per-timestep accuracy but does not ensure that earlier predictions converge toward the full-length prediction, leading to immature or inconsistent results under truncation. Direct alignment with the full-length output is unstable due to the large discrepancy, so TPD aligns cumulative average predictions of consecutive timesteps, providing smoother guidance and ensuring progressive convergence to a unified decision. We compute the running mean logits $\bar{\boldsymbol{z}}_{\leq t}^S$ and probabilities $\bar{\boldsymbol{p}}_{\leq t}^S$ as

$$\bar{\boldsymbol{z}}_{\leq t}^S = \frac{1}{t} \sum_{k=1}^{t} \boldsymbol{z}_k^S, \quad \bar{\boldsymbol{p}}_{\leq t}^S = \text{softmax}_\tau(\bar{\boldsymbol{z}}_{\leq t}^S). \tag{10}$$

TPD encourages the cumulative prediction up to step $t$ to align with that of step $t + 1$, driving predictions to converge progressively over time. Specifically, for every consecutive pair $(t, t + 1)$ with $t \in \{1, \ldots, T - 1\}$, we define

$$\mathcal{L}_{\text{TPD}} = \frac{1}{T - 1} \sum_{t=1}^{T-1} \text{CE}\big(\bar{\boldsymbol{p}}_{\leq t}^S, \bar{\boldsymbol{p}}_{\leq t+1}^S\big) = -\frac{1}{T - 1} \sum_{t=1}^{T-1} \sum_{c=1}^{C} \bar{p}_{\leq t+1,c}^S \log \bar{p}_{\leq t,c}^S. \tag{11}$$

Using the later window as the CE target proved more stable than the KL. In this way, the distillation process progressively transfers information from more stable cumulative averages to earlier, noisier predictions, guiding outputs toward the full-length prediction ("one mind") and ensuring reliable prediction under truncated inference.

**Overall Training Objective.** The overall loss combines the standard classification loss with TMPD and TPD into a single objective:

$$\mathcal{L}_{\text{all}} = \alpha \, \mathcal{L}_{\text{TWD-CE}} + \beta \, \mathcal{L}_{\text{TMPD-KL}} + \gamma \, \mathcal{L}_{\text{TPD}}, \tag{12}$$

where we set $\alpha = 1$, $\beta = 0.5$, and $\gamma = 0.3$.

### 3.3 THEORETICAL ANALYSIS

To theoretically substantiate our findings, we highlight three key properties. First, TMPD introduces an implicit regularization effect on the distillation objective (Theorem 1), which strengthens the temporal supervision by preventing the student from overfitting to individual timestep logits. Second, this stronger supervision allows TMPD to achieve a strictly lower final error floor than TWD, thereby yielding higher ultimate accuracy (Theorem 2). Third, TPD mitigates the accuracy drop under truncated inference, outperforming no-consistency (NC), which ignores temporal alignment, and neighbor-step consistency (NSC), which only matches adjacent steps without cumulative averaging (Theorem 3).

**Theorem 1** (Regularization Effect of TMPD). *TMPD induces an implicit regularization effect on the distillation objective.*

**Theorem 2** (Convergence Robustness Bound of TMPD). *The final error of TMPD, denoted by $\varepsilon_{\text{TMPD}}$, forms a strictly lower bound of $\varepsilon_{\text{TWD}}$ under the same learning rate, as:*

$$\varepsilon_{\text{TMPD}} < \varepsilon_{\text{TWD}}. \tag{13}$$

The proof is provided in Appendix B.1 and B.2, respectively. Together, these results show that TMPD introduces an implicit regularization effect by perturbing the teacher logits at each timestep (Theorem 1). Instead of matching a single deterministic teacher value, the student must remain consistent over a local neighborhood of perturbed teacher signals, resulting in denser and thus stronger temporal supervision. This stronger supervision reduces the correlation of gradient contributions across timesteps (Theorem 2), lowers the effective variance without introducing significant bias, and ultimately leads TMPD to converge to a strictly lower error floor and achieve higher accuracy.

**Theorem 3** (Truncation Robustness Bound of TPD). *Denote by $\Delta_{\text{acc}}(t)$ the truncation accuracy drop at step $t < T$. Then the truncation accuracy drop of TPD is strictly smaller than that of NSC, and NSC is strictly smaller than NC:*

$$\Delta_{\text{acc}}^{\text{TPD}}(t) \; < \; \Delta_{\text{acc}}^{\text{NSC}}(t) \; < \; \Delta_{\text{acc}}^{\text{NC}}(t). \tag{14}$$

The proof is provided in Appendix B.3. For any truncation step $t$, the bound establishes a strict ordering where TPD yields a smaller accuracy drop than NSC, which in turn outperforms NC. This guarantees reliability and higher accuracy under truncated inference. TMPD and TPD provide complementary benefits since TMPD improves full-length accuracy through richer temporal supervision and reduced gradient variance, while TPD enhances truncated inference by exponentially bounding accuracy loss. When combined, the stronger final accuracy achieved by TMPD tightens TPD's bound and yields consistently superior performance across both full-length and truncated timesteps.

## 4 EXPERIMENTS

In this section, we evaluate the effectiveness of our proposed model across different datasets and network architectures, followed by an examination of its temporal flexibility under truncated timesteps. We then conduct ablation studies to analyze the impact of key components, and conclude with an assessment of energy efficiency and supporting visualizations.

### 4.1 EXPERIMENTAL SETTINGS

Experiments are conducted on nodes with AMD EPYC 7742 CPUs (128 cores, 2.25 GHz) and NVIDIA A100 GPUs, with GPU count adjusted according to the experiment scale. LIF neurons with surrogate gradient backpropagation (Huh & Sejnowski, 2018) are implemented using SpikingJelly (Fang et al., 2023). We evaluate the proposed model on ResNet-18/19 and Spiking-Transformer architectures over CIFAR-10/100 and ImageNet. Reported results are averaged over three runs for reliability. Detailed settings are provided in Appendix C.

Table 1: Top-1 accuracy (%) on CIFAR-10/100 with and without KD across different timesteps $T$.

| | METHOD | MODEL | $T = 2$ | $T = 4$ | $T = 6$ |
|---|---|---|---|---|---|
| w/o KD | TET (Deng et al., 2022) | ResNet-19 | 94.16 / 72.87 | 94.44 / 74.47 | 94.50 / 74.72 |
| | STBP-tdBN (Zheng et al., 2021) | ResNet-19 | 92.34 / 69.41 | 92.92 / 70.86 | 93.16 / 71.12 |
| | Dspike (Li et al., 2021) | ResNet-18 | 93.13 / 71.68 | 93.66 / 73.35 | 94.25 / 74.24 |
| | GLIF (Yao et al., 2022) | ResNet-18 | 94.15 / 74.60 | 94.67 / 76.42 | 94.88 / 77.28 |
| | | ResNet-19 | 94.44 / 75.48 | 94.85 / 77.05 | 95.03 / 77.35 |
| | RecDis-SNN (Guo et al., 2022) | ResNet-19 | 93.64 / – | 95.53 / 74.10 | 95.55 / – |
| | RateBP (Yu et al., 2024) | ResNet-18 | 94.75 / 75.97 | 95.61 / 78.26 | 95.90 / 79.02 |
| | | ResNet-19 | 96.23 / 79.87 | 96.26 / 80.71 | 96.36 / 80.83 |
| w/ KD | KDSNN (Xu et al., 2023) | ResNet-18 | – | 93.41 / – | – |
| | Joint A-SNN (Guo et al., 2023) | ResNet-18 | 94.01 / 75.79 | 95.45 / 77.39 | – |
| | | ResNet-34 | 95.13 / 77.11 | 96.07 / 79.76 | – |
| | BKDSNN (Xu et al., 2024b) | ResNet-19 | – | 94.95 / 74.95 | – |
| | HTA-KL (Zhang et al., 2025) | ResNet-19 | **96.68** / 80.51 | 96.76 / 81.03 | – |
| | TWSNN (Yu et al., 2025a) | ResNet-18 | 95.11 / 77.32 | 95.57 / 79.10 | 95.96 / 79.80 |
| | | ResNet-19 | 96.65 / 81.47 | 96.97 / 82.47 | 97.00 / 82.56 |
| | **MEOM (Ours)** | ResNet-18 | **95.51 / 77.99** ±0.06 ± 0.08 | **96.07 / 79.66** ±0.05 ± 0.04 | **96.36 / 80.42** ±0.10 ± 0.06 |
| | | ResNet-19 | 96.65 **/ 81.82** ±0.11 ± 0.05 | **97.13 / 82.85** ±0.07 ± 0.04 | **97.08 / 83.22** ±0.08 ± 0.14 |

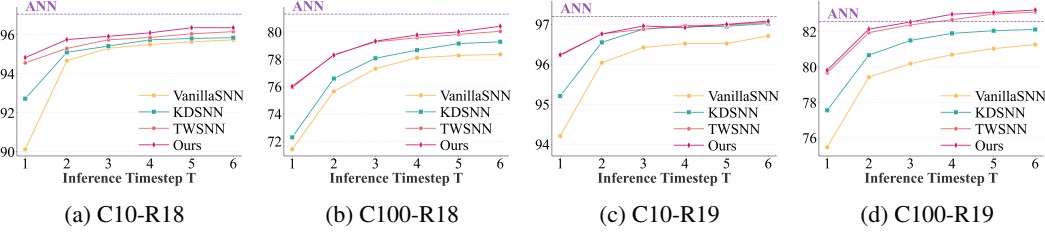

(a) C10-R18  (b) C100-R18  (c) C10-R19  (d) C100-R19

Figure 3: Inference accuracy when models are trained with $T = 6$ and evaluated with truncated timesteps ($T = 1$–5). Here, C10/C100 denote CIFAR-10/100, and R18/R19 denote ResNet-18/19.

## 4.2 PERFORMANCE COMPARISON

For CIFAR-10 and CIFAR-100 (Table 1), MEOM consistently surpasses directly-trained and distillation-based SNN methods across timesteps and model depths. On CIFAR-100, it even slightly outperforms the ANN teacher and achieves clear gains over the strongest SNN baselines, while under low-latency settings ($T = 2$) it still maintains competitive performance. On ImageNet (Table 2), MEOM generalizes effectively across architectures, from convolution-based ResNet-34 to the spiking transformer model, consistently outperforming all existing SNN baselines under the same timestep budget and narrowing the gap to ANN teachers. By overcoming the limitations of a single repeated teacher and isolated timestep training, MEOM provides more informative temporal guidance, enabling the student to learn richer intermediate representations. Combined with consistent gains on both small- and large-scale datasets, this enhanced representational transfer confirms the effectiveness of our method.

## 4.3 TIME FLEXIBILITY

To assess the time flexibility of our approach, we trained all SNNs with a full six-timestep schedule ($T = 6$) on CIFAR-10 and CIFAR-100 using ResNet backbones, and compared against VanillaSNN, KDSNN, and TWSNN. In evaluation, the rollout was progressively truncated from $T = 5$ down to $T = 1$, as shown in Figure 3. MEOM consistently achieved the highest accuracy across all truncation

Table 2: Top-1 accuracy (%) on ImageNet with and without KD. All methods are evaluated at Timestep $T = 4$. S-8-384 denotes the Spiking Transformer architecture (Zhou et al., 2023) with 8 encoder layers and 384-dimensional hidden states.

| | METHOD | MODEL | |
|---|---|---|---|
| | | ResNet-34 | S-8-384 |
| w/o KD | TET (Deng et al., 2022) | 68.00 | – |
| | Dspike (Li et al., 2021) | 68.19 | – |
| | GLIF (Yao et al., 2022) | 67.52 | – |
| | RecDis-SNN (Guo et al., 2022) | 67.33 | – |
| | RateBP (Yu et al., 2024) | 70.01 | – |
| | Spikformer (Zhou et al., 2022) | – | 70.24 |
| | Spikingformer (Zhou et al., 2023) | – | 72.36 |
| | Spike-driven Transformer (Yao et al., 2023) | – | 72.28 |
| w/ KD | KDSNN (Xu et al., 2023) | 67.18 | 74.62 |
| | LaSNN (Hong et al., 2023) | 66.94 | 73.85 |
| | BKDSNN (Xu et al., 2024b) | 67.21 | 75.48 |
| | TWSNN (Yu et al., 2025a) | 71.04 | – |
| | **MEOM (Ours)** | **71.64** | **76.77** |

Table 3: Ablation of components on CIFAR-10/100 with different $T$, with temporal variance measured at $T = 4$ on CIFAR-10.

| TAD | TWD | TMPD | TPD | $T = 2$ | $T = 4$ | $T = 6$ | Var@4 |
|---|---|---|---|---|---|---|---|
| ✗ | ✗ | ✗ | ✗ | 94.67 / 75.16 | 95.00 / 77.22 | 95.74 / 78.37 | 0.493 |
| ✓ | ✗ | ✗ | ✗ | 94.97 / 76.01 | 95.31 / 77.47 | 96.09 / 79.28 | 0.485 |
| ✗ | ✓ | ✗ | ✗ | 95.11 / 77.32 | 95.57 / 79.10 | 96.16 / 79.91 | 0.212 |
| ✗ | ✓ | ✓ | ✗ | 95.43 / 77.78 | 95.98 / 79.58 | 96.31 / 80.13 | 0.133 |
| ✗ | ✓ | ✗ | ✓ | 95.35 / 77.72 | 95.84 / 79.50 | 96.20 / 80.05 | 0.199 |
| ✗ | ✓ | ✓ | ✓ | **95.51 / 77.99** | **96.07 / 79.66** | **96.36 / 80.42** | **0.150** |

points, and on CIFAR-100 with ResNet-19 at $T = 3$ it even surpassed the ANN teacher. At $T = 1$, although accuracy inevitably declined, it still outperformed all baselines. These results demonstrate the effectiveness of TPD in guiding early predictions to converge toward the full-length prediction, thereby enabling reliable prediction under truncated inference.

### 4.4 ABLATION STUDY

**Ablation on TMPD and TPD.** To evaluate the effectiveness of our proposed components, we conduct an ablation study on CIFAR-10 and CIFAR-100 with ResNet-18 with different $T$, as shown in Table 3. Starting from the TAD and TWD baselines, we progressively add TMPD and TPD. Incorporating these components consistently improves accuracy across datasets and timesteps. In addition to improving accuracy, these modules also shape the model's temporal variance, which measures how much the outputs change across timesteps. TMPD reduces variance by smoothing noisy deviations, and TPD lowers variance by improving the consistency of temporal predictions. When used together, the two components preserve meaningful temporal differences while limiting unnecessary variation, and this balanced temporal behavior leads to the highest accuracy in all settings.

**Ablation on Temporal Step Selection in TPD.** Our method, denoted as **Full**, applies temporal consistency across all six timesteps, with the consistency loss computed as the cumulative average over the selected steps. For the reduced-step configurations, we evaluate **R-3** (randomly choosing three timesteps), **F-6-R-2** (always using $t = 6$ plus two random steps), and **F-2-4-6** (using $t = 2, 4, 6$). As shown in Table 4, **Full** achieves the highest accuracy, indicating the benefit of leveraging

Table 4: Comparison of different temporal step selection strategies in TPD.

|          | R-3   | F-6 R-2 | F-2-4-6 | F-6 A-5 | NSC   | **FULL** |
|----------|-------|---------|---------|---------|-------|----------|
| **CIFAR-10**  | 96.19 | 96.24   | 96.14   | 95.99   | 96.19 | **96.36** |
| **CIFAR-100** | 80.30 | 80.35   | 80.16   | 80.11   | 80.36 | **80.42** |

Table 5: Ablation study of hyperparameters $\beta$ and $\gamma$.

| $\beta$ with $\gamma = 0$ | 0.1 | 0.3 | **0.5** | 0.7 | 0.9 |
|---|---|---|---|---|---|
| **ACC (%)** | 95.62 / 79.33 | 95.92 / 79.33 | **95.94 / 79.54** | 95.87 / 79.35 | 95.89 / 79.35 |
| $\gamma$ with $\beta = 0.5$ | 0.1 | **0.3** | 0.5 | 0.7 | 0.9 |
| **ACC (%)** | 95.90 / 79.56 | **96.07 / 79.66** | 95.95 / 79.56 | 95.85 / 79.33 | 95.69 / 79.46 |

all steps. Among the reduced-step variants, **F-6-R-2** performs best, suggesting that coupling the final step with varied earlier ones is more effective than relying on a fixed subset. In addition to these reduced-step variants, we include two broader baselines: **F-6 A-5** (comparing the prediction at $t = 6$ with all five earlier timesteps) and **NSC** (comparing only adjacent timestep pairs). Both **F-6 A-5** and **NSC** perform worse than **Full**, indicating that neither global-only nor local-only comparison is sufficient. Using all timesteps in a progressive manner yields the most stable temporal behavior and the best overall performance.

**Ablation on hyperparameters.** Table 5 reports the hyperparameter ablation under the protocol where we first fix $\alpha = 1$ to select the optimal $\beta$, and then fix this $\beta$ to determine the best $\gamma$. Introducing TMPD with the selected $\beta$ consistently improves accuracy over the TWD baseline, indicating that the hyperparameters produce stable and predictable effects. We also note that the two modules interact with each other, and selecting a compatible set of hyperparameters plays a key role in obtaining the best performance. We adopt $\beta = 0.5$ and $\gamma = 0.3$ in the experiments.

### 4.5 ENERGY EVALUATION AND VISUALIZATION

**Energy Evaluation.** As shown in Table 6, the firing rate (FR) of MEOM is slightly higher than that of the other methods, and its difference from TWSNN is regarded as normal fluctuation. Temporal-wise comparison requires each step to contribute predictive evidence rather than concentrating it at later steps, which shifts some activity earlier and slightly increases the overall FR. Energy is computed from accumulation (ACs) and multiply–accumulate (MACs), and it grows correspondingly with the increase in firing rate. The increase in training time (TT, per batch) and GPU hours mainly comes from TMPD, as it introduces temporally masked teacher logits and multiple KL-divergence computations across timesteps, with no increase in GPU memory, and this remains much cheaper than training multiple teachers. The detailed energy calculation procedure is provided in Appendix D, and additional experiments are shown in Appendix E.

**Visualization.** We utilize t-SNE to visualize features on CIFAR-100, with ResNet-34 as the teacher and ResNet-18 as the student. As shown in Figure 4, our method generates more compact and well-separated clusters compared to prior SNN distillation methods, indicating better feature discriminability and knowledge transfer. In contrast, other methods show more overlapping clusters.

### 5 DISCUSSION

To address the limitations of existing SNN distillation methods that rely on a single teacher signal and overlook full temporal information under truncated inference, we propose MEOM, a unified framework that fully exploits the temporal dimension of SNNs. MEOM integrates TMPD, which introduces diverse teacher supervision to capture multiple temporal perspectives, and TPD, which progressively aligns truncated predictions with the full-length prediction to ensure temporal flex-

Table 6: Comparison of energy consumption, computation operations, training time, GPU hours, GPU memory, and accuracy on CIFAR-10 with ResNet-18 with timestep $T = 6$.

| METHOD | FRs (%) | ACs (M) | MACs (M) | ENERGY ($\mu$J) | TT (ms) | GPU h (h) | GPU Mem (GB) | ACC (%) |
|---|---|---|---|---|---|---|---|---|
| ANN | – | 0.557 | 549.1 | 1757.2 | 31 | 1.07 | 1.99 | 97.06 |
| VanillaSNN | 14.21 | 72.500 | 3.342 | 17.946 | 123 | 4.25 | 11.50 | 95.74 |
| KDSNN | 14.90 | 78.641 | 3.342 | 18.560 | 129 | 4.45 | 11.88 | 95.85 |
| TWSNN | 15.66 | 83.915 | 3.342 | 19.087 | 130 | 4.49 | 11.88 | 96.16 |
| **Ours** | 15.98 | 85.644 | 3.342 | 19.260 | 163 | 5.63 | 11.88 | 96.28 |

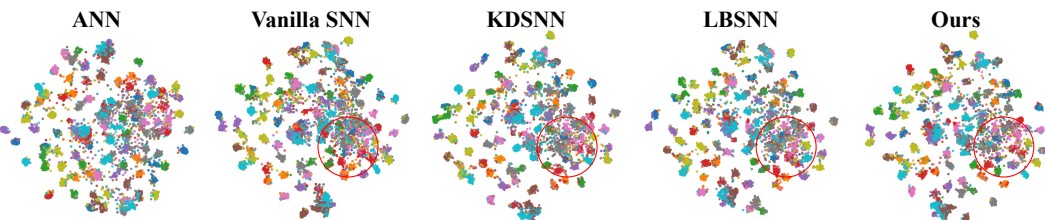

Figure 4: t-SNE visualization of feature representations learned by teacher ANN, vanilla SNN, and various KD methods.

ibility. Extensive theoretical analyses and experiments on multiple benchmarks demonstrate that MEOM substantially improves both accuracy and time flexibility.

**Implications for "one mind".** For the declaration of express, the final full-length output is chosen as the "one mind." In fact, enforcing a single consistent output across timesteps is both a pursuit and a necessity for SNNs, since using fewer timesteps corresponds to lower energy consumption. When the number of timesteps becomes large, accuracy does not always increase monotonically, and earlier timesteps may even outperform the final output. In such cases, the one-mind principle still holds, as the "one mind" can be interpreted as the best output that emerges at an earlier timestep. At the population level, some prefix satisfies $\bar{q}_{1:t^*} = q^\star$. The alignment constraints impose $D(\bar{q}_{1:t}, \bar{q}_{1:t+1}) = 0$, yielding $\bar{q}_{1:1} = \cdots = \bar{q}_{1:T}$. Since one prefix matches $q^\star$, all must, giving $\bar{q}_{1:T} = q^\star$. Thus the final average is Bayes-optimal because any global minimizer collapses all prefix averages to the same optimal predictor.

**Learnable mask for distillation.** Although TMPD diversifies supervision, the masks at each timestep are randomly sampled and therefore do not adapt to the evolving temporal dynamics of spiking sequences. Intuitively, learnable masks could better capture fluctuations in the student's predictions. To evaluate this hypothesis, we compared a learnable parameter mask, a gating mask, and a gumbel-sigmoid mask with random masks, as shown in Table 7. Our results show that random masks consistently achieve the best and most stable performance. We argue that learnable masks tend to adapt to the student's intermediate predictions, which are not yet reliable and therefore introduce biased perturbations. In contrast, simple random masks offer lightweight, unbiased temporal perturbations that better serve the purpose of TMPD. Future work may explore temporally aware models that more effectively capture the underlying temporal variations.

Table 7: Comparison of different mask operations on CIFAR-10 / CIFAR-100.

| Mask Operation | Random (Ours) | Parameter | Gating | Gumbel-Sigmoid |
|---|---|---|---|---|
| **Accuracy (%)** | **96.07 / 79.66** | 95.99 / 79.34 | 95.85 / 79.38 | 95.89 / 79.44 |

ACKNOWLEDGMENTS

The work was supported in part by start-up funds with No. MSRI8001004 and No. MSRI9002005.

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

## A  Structural Temporal Fluctuations in LIF Dynamics

The leak–integrate–reset mechanism of LIF neurons inherently produces temporal fluctuations in membrane potentials, even under constant input. Consider a single LIF neuron receiving a fixed current $I$. Its membrane potential evolves as

$$u_{t+1} = \alpha u_t + I - V_{\text{th}}\, s_t, \qquad s_t \in \{0, 1\}, \tag{15}$$

where $0 < \alpha < 1$ is the leak factor and $V_{\text{th}} > 0$ is the firing threshold. In any non-trivial firing regime (neither always silent nor always firing), the neuron exhibits both spike and non-spike timesteps. Consequently, the update increments take two distinct values:

$$u_{t+1} - u_t = (\alpha - 1)\, u_t + I \qquad \text{(no spike)}, \tag{16}$$

$$u_{t+1} - u_t = (\alpha - 1)\, u_t + I - V_{\text{th}} \qquad \text{(spike)}. \tag{17}$$

These two increments differ by exactly $V_{\text{th}} > 0$, so the sequence $\{u_t\}$ cannot remain constant over time and must visit at least two distinct membrane-potential levels with non-zero frequency. This implies a strictly positive temporal variance, showing that temporal fluctuations arise naturally from the LIF update rule.

## B  Proof of theorems

### B.1  Regularization Effect of TMPD

**Theorem 1.** *TMPD induces an implicit regularization effect on the distillation objective.*

*Proof.* Under TWD, the teacher provides a fixed logit vector $\boldsymbol{z}^A$, and the student minimizes

$$\mathcal{L}_{\text{TWD}} = \ell(\boldsymbol{z}^A), \tag{18}$$

where $\ell(\cdot)$ denotes the distillation loss (KL divergence).

Under TMPD, the teacher logit is perturbed by a temporal mask:

$$\hat{\boldsymbol{z}}^A = \boldsymbol{z}^A \odot (1 + \lambda \boldsymbol{m}) = \boldsymbol{z}^A + \boldsymbol{\varepsilon}, \tag{19}$$

where $\boldsymbol{m}$ is a zero-mean random mask vector and $\boldsymbol{\varepsilon} = \boldsymbol{z}^A \odot (\lambda \boldsymbol{m})$ satisfies $\mathbb{E}[\boldsymbol{\varepsilon}] = \boldsymbol{0}$. Since a new mask is drawn at every iteration, the student minimizes the expected loss

$$\mathcal{L}_{\text{TMPD}} = \mathbb{E}\big[\ell(\boldsymbol{z}^A + \boldsymbol{\varepsilon})\big]. \tag{20}$$

For distillation losses with KL divergence, $\ell(\cdot)$ is convex with respect to the teacher logit vector. Thus, Jensen's inequality yields

$$\ell(\boldsymbol{z}^A) = \ell\big(\mathbb{E}[\boldsymbol{z}^A + \boldsymbol{\varepsilon}]\big) \leq \mathbb{E}\big[\ell(\boldsymbol{z}^A + \boldsymbol{\varepsilon})\big]. \tag{21}$$

Consequently, the TMPD objective can be decomposed as

$$\mathcal{L}_{\text{TMPD}} = \ell(\boldsymbol{z}^A) + \Big(\mathbb{E}\big[\ell(\boldsymbol{z}^A + \boldsymbol{\varepsilon})\big] - \ell(\boldsymbol{z}^A)\Big) = \mathcal{L}_{\text{TWD}} + R(\boldsymbol{z}^A), \tag{22}$$

where the residual term $R(\boldsymbol{z}^A) \geq 0$. This quantity measures the increase in loss under small perturbations of the teacher logits. Minimizing the TMPD objective therefore penalizes solutions that are overly sensitive to such perturbations, which constitutes an implicit regularization effect.

Hence, TMPD can be viewed as optimizing the original TWD loss together with an additional regularization term. □

## B.2 CONVERGENCE ROBUSTNESS BOUND OF TMPD

**Theorem 2.** *The final error of TMPD, denoted by $\varepsilon_{TMPD}$, forms a strictly lower bound of that of TWD, denoted by $\varepsilon_{TWD}$, under the same learning rate, as:*

$$\varepsilon_{TMPD} < \varepsilon_{TWD}. \tag{23}$$

*Proof.* Let $J(\boldsymbol{\theta})$ denote the training objective function parameterized by $\boldsymbol{\theta}$, and let $J^\star$ be its minimum value achieved at the optimal parameter $\boldsymbol{\theta}^\star$. Assume $J(\boldsymbol{\theta})$ is $L$-smooth[1] and satisfies the PL (Polyak–Łojasiewicz) condition[2] with parameter $\mu > 0$.

For SGD with constant step size $\eta \leq 1/L$, the standard bound is

$$\mathbb{E}[J(\boldsymbol{\theta}_k) - J^\star] \leq (1 - \eta\mu)^k (J(\boldsymbol{\theta}_0) - J^\star) + \varepsilon, \tag{24}$$

where $\varepsilon = a\,\bar{\sigma}^2 + b\,\bar{\beta}$, with $a, b > 0$ as coefficients, $\bar{\sigma}^2$ denoting the gradient variance and $\bar{\beta}$ the gradient bias term in standard SGD–PL bounds (Ajalloeian & Stich, 2020; Gower et al., 2021). This shows that the geometric factor $(1 - \eta\mu)^k$ determines the convergence rate, while the asymptotic floor $\varepsilon$ determines the final accuracy. Since TWD and TMPD share the same rate, it suffices to compare $\varepsilon_{\text{TMPD}}$ and $\varepsilon_{\text{TWD}}$.

Let $\boldsymbol{g}_t$ denote the gradient contribution at timestep $t$. For two different timesteps $s$ and $t$, we write $\boldsymbol{g}_s$ and $\boldsymbol{g}_t$. The variance of the time-averaged gradient $\frac{1}{T}\sum_{t=1}^{T}\boldsymbol{g}_t$ decomposes as

$$\text{Var}\left(\frac{1}{T}\sum_{t=1}^{T}\boldsymbol{g}_t\right) = \frac{1}{T^2}\sum_{t=1}^{T}\text{Var}(\boldsymbol{g}_t) + \frac{2}{T^2}\sum_{1 \leq s < t \leq T}\text{Cov}(\boldsymbol{g}_s, \boldsymbol{g}_t). \tag{25}$$

In TWD, all $\boldsymbol{g}_t$ are computed using the same teacher output, which induces positive correlations across timesteps; hence the covariance terms are positive, inflating the effective variance $\bar{\sigma}_{\text{TWD}}^2$.

In TMPD, the teacher outputs are perturbed independently across timesteps, making the cross-time covariances vanish up to $O(\lambda^2)$, while per-timestep variances change by at most $O(\lambda^2)$. Therefore, for some constant $c_v > 0$:

$$\bar{\sigma}_{\text{TMPD}}^2 \leq \bar{\sigma}_{\text{TWD}}^2 - c_v\lambda^2. \tag{26}$$

Since the masks are drawn from a mean-zero distribution, the induced gradient bias is $O(\lambda^2)$ in norm, leading to $\bar{\beta}_{\text{TMPD}} = O(\lambda^4)$, negligible compared to the variance reduction. Thus the error floor satisfies

$$\varepsilon_{\text{TMPD}} = a\,\bar{\sigma}_{\text{TMPD}}^2 + O(\lambda^4) < a\,\bar{\sigma}_{\text{TWD}}^2 + O(\lambda^4) = \varepsilon_{\text{TWD}}. \tag{27}$$

Since both methods share the same geometric factor $(1 - \eta\mu)$, the strictly smaller error floor of TMPD directly implies better final performance than TWD. $\qquad\square$

## B.3 TRUNCATION ROBUSTNESS BOUND OF TPD

**Lemma 1** (Distributional difference bound). *Let $d_{\text{TV}}(p, q) := \frac{1}{2}\|p - q\|_1$ denote the total variation distance. For any sequence of predictive distributions $\{r_0, r_1, \ldots, r_n\}$,*

$$d_{\text{TV}}(r_0, r_n) \leq \sum_{j=0}^{n-1} d_{\text{TV}}(r_j, r_{j+1}) \leq \sqrt{\frac{n}{2}\sum_{j=0}^{n-1}\Phi(r_j, r_{j+1})}, \tag{28}$$

*where $\Phi$ is either the KL divergence or the cross-entropy.*

*Proof.* Since $d_{\text{TV}}$ is a metric, it satisfies the triangle inequality:

$$d_{\text{TV}}(r_0, r_n) \leq \sum_{j=0}^{n-1} d_{\text{TV}}(r_j, r_{j+1}). \tag{29}$$

---

[1]$L$-**smooth:** The gradient is Lipschitz continuous, i.e., $\|\nabla J(\boldsymbol{\theta}_1) - \nabla J(\boldsymbol{\theta}_2)\| \leq L\|\boldsymbol{\theta}_1 - \boldsymbol{\theta}_2\|$.
[2]**PL condition:** A weaker condition than strong convexity, expressed as $\|\nabla J(\boldsymbol{\theta})\|^2 \geq 2\mu(J(\boldsymbol{\theta}) - J^\star)$.

By Pinsker's inequality, for each $j$ we have

$$d_{\text{TV}}(r_j, r_{j+1}) \leq \sqrt{\tfrac{1}{2} D_{\text{KL}}(r_j \| r_{j+1})}. \tag{30}$$

Furthermore, cross-entropy satisfies $H(p,q) = H(p) + D_{\text{KL}}(p\|q)$, so clearly $H(p,q) \geq D_{\text{KL}}(p\|q)$. Hence the same bound holds if we replace KL with cross-entropy:

$$d_{\text{TV}}(r_j, r_{j+1}) \leq \sqrt{\tfrac{1}{2} \Phi(r_j, r_{j+1})}. \tag{31}$$

Summing over $j$ and applying Cauchy–Schwarz then gives

$$\sum_{j=0}^{n-1} d_{\text{TV}}(r_j, r_{j+1}) \leq \sqrt{\tfrac{n}{2} \sum_{j=0}^{n-1} \Phi(r_j, r_{j+1})}. \tag{32}$$

$\square$

**Theorem 3.** *Denote by $\Delta_{acc}(t)$ the truncation accuracy drop at step $t < T$. Then the truncation accuracy drop of TPD is strictly smaller than that of neighbor-step consistency (NSC), and NSC is strictly smaller than no-consistency (NC):*

$$\Delta_{acc}^{TPD}(t) < \Delta_{acc}^{NSC}(t) < \Delta_{acc}^{NC}(t). \tag{33}$$

*Proof.* The truncation accuracy drop at step $t$ is defined as the difference between the accuracy at the final prediction $q_T$ and the truncated prediction $q_t$. The difference in accuracies is upper bounded by the total variation (Devroye et al., 2013):

$$\Delta_{\text{acc}}(t) \leq d_{\text{TV}}(q_t, q_T). \tag{34}$$

Since the truncation accuracy drop at step $t$ is upper bounded by the total variation distance between the truncated distribution $q_t$ and the final distribution $q_T$, it is sufficient to compare the corresponding total variation distances when analyzing truncation errors across different methods.

For **NSC**, the training objective $J_\gamma^{\text{NSC}}(\boldsymbol{\theta})$ is

$$J_\gamma^{\text{NSC}}(\boldsymbol{\theta}) = \text{supervised loss}(\boldsymbol{\theta}) + \gamma \sum_{k=1}^{T-1} \Phi(q_k(\boldsymbol{\theta}), q_{k+1}(\boldsymbol{\theta})), \qquad \gamma > 0. \tag{35}$$

Applying Lemma 1 to the chain $(q_t, \ldots, q_T)$ yields

$$d_{\text{TV}}(q_t, q_T) \leq \sqrt{\tfrac{T}{2} \sum_{k=1}^{T-1} \Phi(q_k, q_{k+1})}. \tag{36}$$

Since the supervised loss is nonnegative, and the penalty term vanishes when the objective is minimized (so that $J_\gamma^{\text{NSC}\star} := \min_{\boldsymbol{\theta}} J_\gamma^{\text{NSC}}(\boldsymbol{\theta})$), we obtain

$$J_\gamma^{\text{NSC}}(\boldsymbol{\theta}) - J_\gamma^{\text{NSC}\star} \geq \gamma \sum_{k=1}^{T-1} \Phi(q_k, q_{k+1}) \implies \sum_{k=1}^{T-1} \Phi(q_k, q_{k+1}) \leq \tfrac{1}{\gamma}\big(J_\gamma^{\text{NSC}}(\boldsymbol{\theta}) - J_\gamma^{\text{NSC}\star}\big). \tag{37}$$

Therefore,

$$d_{\text{TV}}(q_t, q_T) \leq \sqrt{\tfrac{T}{2\gamma}\big(J_\gamma^{\text{NSC}}(\boldsymbol{\theta}) - J_\gamma^{\text{NSC}\star}\big)}. \tag{38}$$

At this point, the truncation error has been linked to the optimization suboptimality $J_\gamma^{\text{NSC}}(\boldsymbol{\theta}) - J_\gamma^{\text{NSC}\star}$. According to Eq. 24, we suppress these terms since our focus is on the geometric convergence rate and the noise contributions are identical across methods. Thus, gradient descent with step size $\eta \leq 1/L$ ensures

$$J_\gamma^{\text{NSC}}(\boldsymbol{\theta}_m) - J_\gamma^{\text{NSC}\star} \leq (1 - \eta\mu_\gamma)^m \big(J_\gamma^{\text{NSC}}(\boldsymbol{\theta}_0) - J_\gamma^{\text{NSC}\star}\big). \tag{39}$$

Combining the inequalities shows that the truncation error in NSC decays geometrically:

$$\Delta_{\text{acc}}^{\text{NSC}}(t) \ \leq \ \sqrt{\tfrac{T}{2\gamma}} \ \sqrt{(1 - \eta\mu_\gamma)^m \, (J_\gamma^{\text{NSC}}(\boldsymbol{\theta}_0) - J_\gamma^{\text{NSC}\star})}. \tag{40}$$

For **TPD**, we define averaged predictions $\bar{q}_{\leq k} = \tfrac{1}{k}\sum_{i=1}^{k} q_i$. The training objective is

$$J_\gamma^{\text{TPD}}(\boldsymbol{\theta}) \ = \ \text{supervised loss}(\boldsymbol{\theta}) \ + \ \gamma \sum_{k=1}^{T-1} \Phi(\bar{q}_{\leq k}(\boldsymbol{\theta}), \, \bar{q}_{\leq k+1}(\boldsymbol{\theta})) \, . \tag{41}$$

Applying Lemma 1 to the averaged chain $(\bar{q}_{\leq t}, \dots, \bar{q}_{\leq T})$ gives

$$d_{\text{TV}}(\bar{q}_{\leq t}, \bar{q}_{\leq T}) \ \leq \ \sqrt{\tfrac{T}{2\gamma} \left( J_\gamma^{\text{TPD}}(\boldsymbol{\theta}) - J_\gamma^{\text{TPD}\star} \right)}. \tag{42}$$

By applying the same reasoning as in the NSC case above, the truncation error also decays geometrically:

$$\Delta_{\text{acc}}^{\text{TPD}}(t) \ \leq \ \sqrt{\tfrac{T}{2\gamma}} \ \sqrt{(1 - \eta\tilde{\mu}_\gamma)^m \, (J_\gamma^{\text{TPD}}(\boldsymbol{\theta}_0) - J_\gamma^{\text{TPD}\star})}. \tag{43}$$

Temporal averaging reduces variance across steps, smoothing the discrepancy signal and increasing the PL constant, so that $\tilde{\mu}_\gamma \geq \mu_\gamma$. Consequently, the truncation error in TPD decays strictly faster than in NSC, yielding

$$\Delta_{\text{acc}}^{\text{TPD}}(t) \ < \ \Delta_{\text{acc}}^{\text{NSC}}(t). \tag{44}$$

For **NC**, the objective reduces to

$$J^{\text{NC}}(\boldsymbol{\theta}) = \text{supervised loss}(\boldsymbol{\theta}). \tag{45}$$

However, unlike in NSC or TPD, the sum of divergences $\sum_k \Phi(q_k, q_{k+1})$ is not penalized in the objective and hence not reduced during optimization. These discrepancies may remain large even as the supervised loss decreases, so the truncation error does not benefit from geometric decay. As a result, the error in NC is strictly larger than in NSC:

$$\Delta_{\text{acc}}^{\text{NSC}}(t) \ < \ \Delta_{\text{acc}}^{\text{NC}}(t). \tag{46}$$

Putting the three cases together, we obtain the strict ordering

$$\Delta_{\text{acc}}^{\text{TPD}}(t) \ < \ \Delta_{\text{acc}}^{\text{NSC}}(t) \ < \ \Delta_{\text{acc}}^{\text{NC}}(t). \tag{47}$$

Moreover, each divergence term $\Phi(q_k, q_{k+1})$ arises from stochastic parameter updates, and its size is ultimately controlled by the gradient variance and bias that also determine the error floor in TMPD.

Finally, recall that the temporal error at step $t$ decomposes as

$$\varepsilon(t) \ = \ \varepsilon_{\text{full}} + \Delta_{\text{acc}}(t). \tag{48}$$

TMPD reduces the full-length error term $\varepsilon_{\text{full}}$ by lowering the variance of the teacher signal (Theorem 2), while TPD reduces the truncation term $\Delta_{\text{acc}}(t)$ by tightening the temporal alignment (Theorem 3). Since these two quantities enter the error additively and affect disjoint parts of the decomposition, their benefits are complementary rather than redundant.

Thus, when TMPD and TPD are combined in MEOM, the error at step $t$ can therefore be written as

$$\begin{aligned} \varepsilon^{\text{MEOM}}(t) \ &= \ \varepsilon_{\text{full}}^{\text{TMPD}} \ + \ \Delta_{\text{acc}}^{\text{TPD}}(t) \\ &< \ \varepsilon_{\text{full}}^{\text{TWD}} \ + \ \Delta_{\text{acc}}^{\text{NC}}(t) \ = \ \varepsilon^{\text{baseline}}(t), \end{aligned} \tag{49}$$

which shows that MEOM simultaneously reduces the full-length error term (through TMPD) and the truncation term (through TPD). In this sense, the two components act on complementary parts of the temporal error decomposition and their effects add up, yielding uniformly better performance across timesteps compared with using either component alone.

$$\square$$

## C    EXPERIMENTAL SETTINGS

**Dataset.**    CIFAR-10 and CIFAR-100 are benchmark datasets for image classification. CIFAR-10 consists of 10 classes with 60,000 32 × 32 color images, divided into 50,000 training and 10,000 testing samples. CIFAR-100 extends this to 100 classes grouped into 20 superclasses, with the same total number of images but fewer per class, making it more challenging. Direct encoding is utilized to convert image pixels into time series, with pixel values repeatedly fed into the input layer at each timestep.

ImageNet is a large-scale image classification dataset containing over 1.2 million training images and 50,000 validation images spanning 1,000 object categories. Each image varies in resolution and visual complexity, providing a more challenging and diverse benchmark compared to CIFAR datasets. To align with the experimental setup, images are resized and center-cropped to a resolution of 224 × 224 before being processed into time series representations through direct encoding.

**Experiment Details.**    All models are trained using stochastic gradient descent (SGD) with a momentum of 0.9, combined with a cosine annealing learning rate schedule. The experiments are implemented in PyTorch. For the CIFAR datasets, training is conducted on a single NVIDIA A100 GPU, while ImageNet experiments employ distributed data parallel training across eight A100 GPUs to maximize computational efficiency, accelerate convergence, and ensure training stability. Different student architectures are paired with their corresponding teacher models for training, and all hyperparameter settings and network architectures are summarized in Table 8.

Table 8: Training settings and architecture across datasets. Here, **LR** denotes the learning rate, and **WD** denotes the weight decay.

| DATASET | BATCH SIZE | EPOCHS | LR | WD | STUDENT ARCH. | TEACHER ARCH. | TEACHER ACC(%) |
|---|---|---|---|---|---|---|---|
| CIFAR-10 | 128 | 300 | 0.1 | 5e-4 | ResNet-18 ResNet-19 | ResNet-34 ResNet-19 | 97.06 97.20 |
| CIFAR-100 | 128 | 300 | 0.1 | 5e-4 | ResNet-18 ResNet-19 | ResNet-34 ResNet-19 | 81.31 82.57 |
| IMAGENET | 512 64 | 100 300 | 0.2 0.1 | 2e-5 5e-2 | ResNet-34 S-8-384 | ResNet-34 ViT-Base | 71.24 81.78 |

## D    ENERGY CONSUMPTION ANALYSIS

To evaluate the energy efficiency of SNNs, we adopt a standard neuromorphic computing methodology that estimates the total synaptic operation power (SOP) based on the number of fundamental operations and their associated energy costs (Zhou et al., 2022). The SOP is defined as:

$$SOP_s = E_{\text{AC}} \cdot AC_s + E_{\text{MAC}} \cdot MAC_s, \tag{50}$$

where $E_{\text{AC}}$ and $E_{\text{MAC}}$ represent the energy consumption per accumulation (AC) and per multiply-accumulate (MAC) operation, respectively. Following the energy model in (Han et al., 2015), we assume that each 32-bit floating-point addition consumes $0.9$ picojoules (pJ), while each MAC operation consumes $4.6$ pJ.

In SNNs, neurons transmit binary spike signals, $s_i^l[t] \in \{0, 1\}$, indicating whether neuron $i$ in layer $l$ fires at timestep $t$. A firing spike activates all its outgoing synapses, with each synapse performing an addition. If a neuron has $f_i^l$ outgoing connections (fan-out), the total number of AC operations across the network can be expressed as:

$$AC_s = \sum_{t=1}^{T} \sum_{l=1}^{L-1} \sum_{i=1}^{N^l} f_i^l \cdot s_i^l[t], \tag{51}$$

where $T$ is the total number of timesteps, $L$ is the number of layers, and $N^l$ denotes the number of neurons in layer $l$.

In contrast, ANNs operate without temporal dynamics. Each neuron performs a single forward pass, requiring a fixed number of MAC operations determined by its synaptic connections:

$$MAC_s = \sum_{l=1}^{L-1} \sum_{i=1}^{N^l} f_i^l. \tag{52}$$

By combining the counts of AC and MAC operations with their respective energy costs, the total SOP for any given network configuration can be effectively estimated.

## E    ENERGY EVALUATION

Table 9 presents the comparison of energy consumption, computation, training time, and accuracy across different methods. Energy is computed from accumulation operations (ACs) and multiply–accumulate operations (MACs), where MACs remain identical because each LIF neuron applies a fixed leak-factor multiplication at every timestep, independent of spiking input. Thus, the main variation comes from ACs, which are directly determined by the firing rate (FR). Compared with VanillaSNN and KDSNN, both TWSNN and our method exhibit higher FR. This arises from the temporal-wise comparison mechanism, which forces the network to increase responsiveness at earlier timesteps, rather than relying predominantly on later steps to accumulate predictive evidence. By shifting part of the activity forward, the average FR rises, leading to correspondingly higher ACs and slightly greater energy. In terms of training time (TT, measured per batch), our method shows a moderate increase compared with other SNNs, mainly due to the mask-weighted perturbed teacher logits introduced in TMPD. At each timestep, TMPD requires generating perturbed logits and computing the corresponding distillation losses, which adds extra overhead during training. However, this additional cost only occurs in the training phase and does not affect inference latency or energy efficiency, making it limited and acceptable.

Table 9: Comparison of energy consumption, computation operations, training time, GPU hours, and accuracy on CIFAR-10 and CIFAR-100 with ResNet-18 and ResNet-19 at timestep $T = 6$.

| MODEL | METHOD | FR (%) | ACs (M) | MACs (M) | ENERGY ($\mu$J) | TT (ms) | GPU h (h) | ACC (%) |
|---|---|---|---|---|---|---|---|---|
| | ANN | – | 0.557 | 549.1 | 1757.3 | 31 | 1.07 | 97.06 |
| | VanillaSNN | 14.21 | 72.500 | 3.342 | 17.946 | 123 | 4.25 | 95.74 |
| C10-R18 | KDSNN | 14.90 | 78.641 | 3.342 | 18.560 | 129 | 4.45 | 95.85 |
| | TWSNN | 15.66 | 83.915 | 3.342 | 19.087 | 130 | 4.49 | 96.16 |
| | **Ours** | 15.98 | 85.644 | 3.342 | 19.260 | 163 | 5.63 | 96.28 |
| | ANN | – | 1.442 | 2268.6 | 7259.7 | 32 | 1.10 | 97.20 |
| | VanillaSNN | 13.06 | 285.764 | 8.652 | 56.264 | 308 | 10.64 | 96.71 |
| C10-R19 | KDSNN | 12.30 | 272.512 | 8.652 | 54.939 | 317 | 10.95 | 97.02 |
| | TWSNN | 15.35 | 310.799 | 8.652 | 58.767 | 318 | 10.98 | 97.05 |
| | **Ours** | 15.20 | 311.080 | 8.652 | 58.796 | 363 | 12.54 | 97.08 |
| | ANN | – | 0.557 | 549.2 | 1757.4 | 26 | 0.90 | 81.31 |
| | VanillaSNN | 17.33 | 93.829 | 3.342 | 20.080 | 128 | 4.42 | 78.37 |
| C100-R18 | KDSNN | 17.80 | 96.175 | 3.342 | 20.315 | 135 | 4.66 | 79.28 |
| | TWSNN | 18.83 | 103.222 | 3.342 | 21.020 | 135 | 4.66 | 80.05 |
| | **Ours** | 17.68 | 97.703 | 3.342 | 20.468 | 170 | 5.87 | 80.42 |
| | ANN | – | 1.442 | 2268.6 | 7259.7 | 235 | 8.11 | 82.57 |
| | VanillaSNN | 16.08 | 350.697 | 8.653 | 62.759 | 324 | 11.19 | 81.27 |
| C100-R19 | KDSNN | 16.32 | 359.646 | 8.653 | 63.654 | 332 | 11.47 | 82.12 |
| | TWSNN | 17.33 | 369.744 | 8.653 | 64.664 | 333 | 11.50 | 83.10 |
| | **Ours** | 17.11 | 366.015 | 8.653 | 64.291 | 381 | 13.17 | 83.22 |

# F    ABLATION STUDY ON IMAGENET

To assess the contribution of each component on a large-scale dataset, we conduct an ablation study on ImageNet using the ResNet-34 SNN backbone with $T = 4$. As shown in Table 10, incorporating TMPD and TPD yields additional improvements over the TWD baseline, and the full configuration achieves the highest accuracy.

Table 10: Ablation of components on ImageNet under the ResNet-34 SNN backbone ($T = 4$).

| TAD | TWD | TMPD | TPD | ACC (%) |
|-----|-----|------|-----|---------|
| ✗ | ✗ | ✗ | ✗ | 66.93 |
| ✓ | ✗ | ✗ | ✗ | 68.22 |
| ✗ | ✓ | ✗ | ✗ | 71.04 |
| ✗ | ✓ | ✓ | ✗ | 71.55 |
| ✗ | ✓ | ✗ | ✓ | 71.34 |
| ✗ | ✓ | ✓ | ✓ | **71.64** |

# G    USE OF LARGE LANGUAGE MODELS

We used LLMs only to polish the writing (e.g., grammar and readability). All ideas, experiments, and analyses are entirely the authors' own.

