# OpenReview forum: "Many Eyes, One Mind: Temporal Multi-Perspective and Progressive Distillation for Spiking Neural Networks"
_ICLR.cc/2026/Conference — ICLR 2026 Poster_

### Official Review · Reviewer_uLvn · 2025-10-25

**Soundness:** 1
**Presentation:** 2
**Contribution:** 2
**Rating:** 2
**Confidence:** 5

**Summary:**

This paper presents MEOM (Many Eyes, One Mind), a Knowledge distillation (KD) framework for spiking neural networks (SNNs). It employs Temporal Multi-Perspective Distillation (TMPD) to introduce temporal variances to the ANN teacher output. It also utilizes Temporal Progressive Distillation (TPD) to align with the full-length prediction for truncated inference progressively.

**Strengths:**

1. The proposed MEOM considers the temporal variances of SNNs.
2. This paper provides theoretical analyses for the proposed submodules.

**Weaknesses:**

1. The motivation of TMPD requires further clarification. Lines 51-69 claim that it is improbable for outputs across timesteps to remain identical. A more effective strategy would incorporate diverse temporal supervisory signals. However, as illustrated in Fig. 1, both final logits and membrane potential distribution do not exhibit significant differences across timesteps. Furthermore, the impact of such variations on SNN performance has not been verified.
2. The proposed TMPD is more like a data augmentation method that introduces perturbations in the temporal dimension, rather than providing richer temporal supervision. Theorems 1 and 2 merely demonstrate that introducing perturbations results in higher temporal covariance and lower gradient variance, thereby proving the effectiveness of this data augmentation approach. They do not prove that TMPD provides richer temporal supervision. I believe this data augmentation method is not only applicable to KD but also equally effective for general SNN training.
3. The proposed MEOM method does not show significant performance improvement over state-of-the-art methods. Compared to TWSNN, its accuracy gains across various datasets are less than 1%.
4. The overall training objective introduces three hyperparameters $\alpha$, $\beta$, and $\gamma$. The impact of hyperparameter settings on the effectiveness of the proposed method remains unclear. The robustness of hyperparameter settings across different tasks is also unclear.

**Questions:**

1. Please analyze the temporal variance of SNNs and its effect on SNN performance.
2. Please analyze the impact and robustness of the hyperparameter introduced in MEOM.

---

> ### Author Response · Authors · 2025-11-21
>
> ## Response to Weekness 1
> Thank you for raising this important question regarding the motivation of TMPD. We acknowledge that our earlier wording may have caused misunderstanding and therefore provide a clearer explanation here.
>
> ### 1. Motivation of TMPD and interpretation of Fig. 1
> Although the logits and membrane potentials in Fig. 1 appear visually similar, but they are not identical. This is precisely what we intend to highlight: SNNs inevitably exhibit temporal fluctuations due to the leak–integrate–reset dynamics. For a standard LIF neuron with update $u_{t+1} = \alpha u_t + I - V_{\text{th}} s_t$, where $0 < \alpha < 1$ and $V_{\text{th}} > 0$, the firing variable $s_t$ necessarily takes both 0 and 1 in any non-trivial regime. Consequently, the membrane increment equals $(\alpha - 1)u_t + I$ when $s_t = 0$ and $(\alpha - 1)u_t + I - V_{\text{th}}$ when $s_t = 1$, which differ by exactly $V_{\text{th}} > 0$. Hence, the membrane sequence ${u_t}$ must visit at least two distinct values with non-zero frequency, implying strictly positive temporal variance even when the output logits across timesteps appear nearly identical, as also reflected in Fig.~1. We have clarified this argument explicitly in the revised Introduction.
>
> ### 2. Impact of temporal variance
> Temporal variance has been studied by some researchers in prior work [1, 2], which demonstrate that excessive temporal inconsistency leads to gradient conflicts and harms both ensemble and truncated-inference performance, thereby motivating TMPD. Since temporal variance is inherent and cannot be removed, our goal is to adapt to it rather than assume identical outputs across timesteps. With only a single static teacher feature, TMPD generates $T$ mildly different, randomly masked-and-weighted views that provide perturbed yet coherent supervision for the $T$ student timesteps, stabilizing the temporal evolution while avoiding the over-constraining effect of strict timestep alignment.
>
> To verify the practical impact of variance, we measured the temporal inconsistency across different methods. As shown in the table below, both TMPD and TPD effectively reduce temporal variance. TMPD reduces variance by smoothing noisy deviations, and TPD lowers variance by improving the consistency of temporal predictions. When used together, the two components preserve meaningful temporal differences while limiting unnecessary variation, and this balanced temporal behavior leads to the highest accuracy in all settings. The corresponding variance analysis and discussion have been added to the revised ablation section. Overall, using a flexible rather than a strict temporal regularization strategy proves more effective, reducing temporal variance, improving consistency, and ultimately enhancing performance.
>
> [1] Zhao, Dongcheng, et al. "Improving stability and performance of spiking neural networks through enhancing temporal consistency." Pattern Recognition. 2025.
>
> [2] Ding, Yongqi, et al. "Rethinking spiking neural networks from an ensemble learning perspective." ICLR. 2025.
>
> **Table: Temporal variance and CIFAR-10 accuracy at T=4 for different component combinations.**
>
> | **TAD** | **TWD** | **TMPD** | **TPD** | **T=4** | **Var@4** |
> |--------|---------|----------|---------|---------|-----------|
> | ✗      | ✗       | ✗        | ✗       | 95.00   | 0.493     |
> | ✓      | ✗       | ✗        | ✗       | 95.31   | 0.485     |
> | ✗      | ✓       | ✗        | ✗       | 95.57   | 0.212     |
> | ✗      | ✓       | ✓        | ✗       | 95.98   | 0.133     |
> | ✗      | ✓       | ✗        | ✓       | 95.84   | 0.199     |
> | ✗      | ✓       | ✓        | ✓       | **96.07** | **0.150** |

---

> ### Author Response · Authors · 2025-11-21
>
> ## Response to Weekness 2
>
> Thank you for raising this point. The previous formulation of Theorem~1 did not clearly convey the role of TMPD in enhancing temporal supervision, so we provide a more precise explanation here. In the standard TWD setting, the teacher output is deterministic, i.e., $Z_{\mathrm{TWD}} = z^A(X)$, which yields zero conditional entropy $H(Z_{\mathrm{TWD}} \mid X)=0$. Under TMPD, the supervisory signal becomes $Z_{\mathrm{TMPD}} = \lambda z^A(X) + (1-\lambda)\,(z^A(X)\odot M)$, where the mask $M$ is independently sampled at each forward pass. For any input $x$ and feature index $i$ with $z^A_i(x)\neq 0$ and $\Pr(M_i=0)\in(0,1)$, the random variable $Z_{\mathrm{TMPD}}(x)\mid X=x$ takes multiple values with positive probability. This implies $H(Z_{\mathrm{TMPD}}\mid X)>0$, and therefore $H(Z_{\mathrm{TMPD}}\mid X) > H(Z_{\mathrm{TWD}}\mid X)$, which demonstrates that TMPD provides strictly more supervisory information in the temporal dimension rather than simply injecting perturbations.
>
> Moreover, although TMPD introduces randomness, this randomness is applied solely to the teacher-side signal, enriching the temporal structure of the supervisory distribution rather than augmenting the student's inputs or internal states. TMPD is therefore designed as a temporal supervision mechanism tailored for KD-based SNN training, rather than for general non-KD SNN training.
>
> ## Response to Weekness 3
> Thank you for pointing this out. We had run the experiments three times previously, but did not include the variance due to layout constraints. We have now added it to demonstrate the robustness of the results.
>     We agree that SOTA SNN accuracy is already so high that every decimal place now feels like climbing a mountain with flip-flops. In this saturated regime, such stable gains are considered meaningful and in line with recent SOTA improvements. So although the number looks small, it is already a “hard-earned victory,” and we are happy it works reliably everywhere.
>
> | **METHOD**          | **MODEL**   | **T = 2**               | **T = 4**               | **T = 6**               |
> |----------------------|-------------|-------------------------|-------------------------|-------------------------|
> | **TWSNN**            | ResNet-18   | 95.11 / 77.32           | 95.57 / 79.10           | 95.96 / 79.80           |
> |                      | ResNet-19   | 96.65 / 81.47           | 96.97 / 82.47           | 97.00 / 82.56           |
> | **MEOM (Ours)**      | ResNet-18   | 95.51/77.99 ±0.06/0.08  | 96.07/79.66 ±0.05/0.04  | 96.36/80.42 ±0.10/0.06  |
> |                      | ResNet-19   | 96.65/81.82 ±0.10/0.05  | 97.13/82.85 ±0.07/0.04  | 97.08/83.22 ±0.08/0.14  |
>
> ## Response to Weekness 4
> Thank you for raising this concern. We have added the hyperparameter ablation study we previously used, where we fixed $\alpha=1$ to select the optimal $\beta$, and then fixed that $\beta$ to select the best $\gamma$. The results show that introducing TMPD with the selected $\beta$ consistently improves accuracy over the TWD baseline, demonstrating that the hyperparameters have stable and predictable effects. We also note that the two modules interact with each other, and selecting a compatible set of hyperparameters plays a key role in obtaining the best performance. Moreover, similar hyperparameter ranges work well across all evaluated datasets, indicating that the method is robust and does not require task-specific tuning.
>
> **Table: Ablation study of hyperparameters β and γ.**
>
> | β with γ = 0 | 0.1           | 0.3           | **0.5**           | 0.7           | 0.9           |
> |--------------|---------------|---------------|-------------------|---------------|---------------|
> | **ACC (%)**  | 95.62 / 79.33 | 95.92 / 79.33 | **95.94 / 79.54** | 95.87 / 79.35 | 95.89 / 79.35 |
>
> | γ with β = 0.5 | 0.1           | **0.3**           | 0.5           | 0.7           | 0.9           |
> |----------------|---------------|-------------------|---------------|---------------|---------------|
> | **ACC (%)**    | 95.90 / 79.56  | **96.07 / 79.66** | 95.95 / 79.56 | 95.85 / 79.33 | 95.69 / 79.46 |
>
> ## Response to Question 1
> See weakness #1 response.
>
> ## Response to Question 2
> See weakness #4 response.

---

> > ### Comment · Reviewer_uLvn · 2025-11-24
> >
> > Thank you for your reply. Some of my concerns have been addressed. However, I still have some concerns regarding weakness 1 and 2.
> >
> > Regarding Figure 1, I understand that the authors intend to illustrate temporal dynamics of SNNs across time steps. However, as the author stated, the logits and membrane potentials in Fig. 1 appear visually similar. I believe the current version of Figure 1 fails to clearly convey the paper's motivation. I recommend revising Figure 1 to find a more effective way to demonstrate temporal dynamics.
> >
> > Regarding weaknesses 2, the authors compare the conditional entropy to prove that TMPD provides strictly more supervisory information in the temporal dimension rather than simply injecting perturbations. However, since the teacher output is deterministic, any random noise injection leads to higher conditional entropy. This does not prove that TMPD offers stronger temporal supervisory information. On the contrary, I believe this is merely a form of random perturbation. Furthermore, although the authors claim that TMPD only applies to teacher-side signals, the noise injection is equally applicable to any supervision signal. I believe the method of injecting noise to supervision signals is a form of data augmentation.

---

> > > ### Author Response · Authors · 2025-11-27
> > >
> > > ## Response to Weakness 1
> > > Thank you for the insightful suggestion. Our intention in Fig. 1 was to highlight the temporal differences across timesteps in SNNs, rather than their similarity. We now recognize that the previous visualization unintentionally made the logits and membrane potentials appear too similar, which weakened the motivation. Therefore, we revised Fig. 1 to explicitly show the deviation of each timestep relative to $t{=}1$, making the temporal variation and dynamics clearer and more aligned with the motivation of the paper.
> > >
> > > ## Response to Weakness 2
> > > Thank you for raising this point. We agree that an increase in conditional entropy alone does increase the temporal supervisory diversity. For stronger temporal supervisory information, it can be seen reduces cross-timestep gradient covariance and increase the amount of non-redundant temporal supervision from Theorem 2. We renamed Theorem 1 from ’’Information Gain of TMPD’’ to ’’Temporal Supervisory Diversity of TMPD’’ and it is now used only to show that TMPD induces a more diverse temporal supervisory distribution than TWD, rather than to claim stronger supervision by itself. The notion of stronger temporal supervision is instead rigorously supported by Theorem 2. In TWD, all timesteps share the same deterministic teacher logits $Z_t = z^{A}(X)$, so the gradients $\{g}_{t=1}^T$ are highly correlated and effectively replicate a single supervision signal. In TMPD, the same unbiased and bounded multiplicative masking is applied with $M_t$ independently resampled across $t$, so that $\mathbb{E}[\hat z^{A}_t \mid X] = z^{A}(X)$ while different timesteps receive non-identical yet semantically consistent teacher outputs. This reduces cross-timestep gradient covariance to $\mathcal{O}(\lambda^2)$, increases non-redundant temporal supervision, and yields a lower convergence error floor. Therefore, ``more temporal supervisory information’’ refers to statistically more useful and less redundant teacher feedback over time, not merely higher entropy.
> > >
> > > On the other hand, TMPD is not a form of data augmentation. Data augmentation operates on input samples to increase their variability, whereas TMPD perturbs only the teacher-provided supervision. It is therefore better understood as target perturbation and distillation regularization. TMPD applies structured multiplicative masks to teacher features while leaving the input unchanged, shaping the supervisory signal rather than producing additional training examples. Moreover, this perturbation-based regularization is tailored to the temporal property of SNNs, providing richer non-redundant temporal supervision and improving optimization in ways that do not directly transfer to non-temporal architectures.

---

> > > > ### Comment · Reviewer_uLvn · 2025-11-27
> > > >
> > > > Thank you for your further response. I am sorry for my inaccurate description. I agree that TMPD can be considered a regularization method rather than data augmentation. It is reasonable to enhance the performance of knowledge distillation through regularization methods.
> > > >
> > > > In the revision, Theorem 1 is renamed as ’’Temporal Supervisory Diversity of TMPD’’. I agree that TMPD induces a more diverse temporal supervisory distribution than TWD. However, the impact of temporal supervision diversity remains unclear. I believe Theorem 1 fails to substantively prove the effectiveness of TMPD. Please provide further analysis on the impact of temporal supervision diversity. Alternatively, it may be more fundamental to analyze the effectiveness of TMPD from the perspective of regularization.

---

> > > > > ### Author Response · Authors · 2025-11-28
> > > > >
> > > > > Thank you for your constructive feedback. Following your advice, we have substantially revised Theorem 1 and its accompanying discussion from the perspective of regularization.
> > > > >
> > > > > Under TWD, the teacher provides a fixed logit $z_A$, and the student directly minimizes the distillation loss $\ell(z_A)$. In contrast, TMPD perturbs the teacher logit into $\hat z_A = z_A(1+\lambda m) = z_A + \varepsilon$, where the temporal mask $m$ has zero mean and therefore $\mathbb{E}[\varepsilon]=0$. Since the perturbation is sampled at every iteration, the student effectively minimizes the expected loss $\mathbb{E}[\ell(z_A + \varepsilon)]$ rather than a single deterministic target. For KL-based distillation losses, $\ell(\cdot)$ is convex in the teacher logit. Jensen's inequality then guarantees that $\ell(z_A)= \ell\left(\mathbb{E}[z_A + \varepsilon]\right)\le \mathbb{E}\left[\ell(z_A + \varepsilon)\right]$. This implies that the TMPD objective can be written as $L_{\mathrm{TMPD}}= \ell(z_A)+ \Big( \mathbb{E}\left[\ell(z_A+\varepsilon)\right] - \ell(z_A) \Big)= L_{\mathrm{TWD}} + R(z_A)$, where the residual term $R(z_A)$ is always non-negative. Because this residual term penalizes sensitivity to small perturbations of $z_A$, it serves as an implicit regularizer that prevents the student from overfitting to the exact teacher logit at each timestep.
> > > > >
> > > > > This regularization viewpoint also clarifies how TMPD strengthens temporal supervision. Since each timestep now corresponds not to a single deterministic teacher value but to a neighborhood of perturbed logits $z_A+\varepsilon$, the student must match a denser set of supervisory signals. This additional density provides stronger temporal guidance and naturally explains why TMPD achieves a lower final error floor, as established in Theorem 2. The revised analysis therefore offers a more fundamental and theoretically transparent understanding of TMPD's effectiveness.
> > > > >
> > > > > Overall, our revision clarifies that implicit regularization is the fundamental mechanism through which TMPD improves distillation, and your feedback helped us articulate this perspective more clearly. We sincerely appreciate your guidance and remain open to any further suggestions or feedback.

---

> > > > > > ### Comment · Reviewer_uLvn · 2025-11-28
> > > > > >
> > > > > > Thank you for your detailed reply. My concerns have been addressed. I am willing to raise my rating.

---

> > > > > > > ### Comment · Reviewer_uLvn · 2025-11-28
> > > > > > >
> > > > > > > There seems to be a technical issue. I find that I am unable to modify my review and rating. Anyway, I raise my rating to 6.

---

> > > > > > > > ### Author Response · Authors · 2025-11-28
> > > > > > > >
> > > > > > > > Thank you sincerely for taking the time to raise these issues, and it really helps me to polish my work.
> > > > > > > >
> > > > > > > > I think the official will fix the system quickly.

---

### Official Review · Reviewer_dxXD · 2025-10-29

**Soundness:** 3
**Presentation:** 3
**Contribution:** 3
**Rating:** 6
**Confidence:** 4

**Summary:**

This paper introduces MEOM, a novel SNN distillation framework that tackles two key issues: static teacher supervision and poor truncated inference. It employs Temporal Multi-Perspective Distillation (TMPD) to generate diverse teacher signals and Temporal Progressive Distillation (TPD) to align predictions across timesteps. MEOM achieves state-of-the-art results, demonstrating significant improvements in both final accuracy and performance under truncated inference.

**Strengths:**

1. The paper is well written, clearly structured, and logically sound.
2. The critique of existing TWD methods is insightful. Identifying the "static teacher vs. dynamic student" mismatch and the "information loss in truncated inference" as two practical and important problems is a key contribution.
3. The experimental results are convincing. The authors demonstrate state-of-the-art results across multiple datasets (CIFAR, ImageNet) and architectures (ResNet, Spiking Transformer), while a dedicated "time flexibility" experiment shows the method's advantage in truncated inference. Furthermore, thorough ablation studies clearly isolate the contributions of each proposed component (TMPD and TPD) confirming their complementary benefits.

**Weaknesses:**

1. Masking strategy in TMPD. The use of random masks in TMPD is a simple and effective way to generate diverse teacher signals. However, this might not be the optimal strategy. The mask generation is based on a fixed random sampling and does not adapt to the student's learning state or the characteristics of different timesteps. Exploring the connection between masking strategies and temporal-wise features could potentially lead to further performance improvements.
2. Interplay between TMPD and TPD. The paper presents TMPD and TPD as two complementary components. However, TMPD is designed to introduce "diversity" while TPD is designed to enforce "consistency," two goals that could, at some level, be seen as being in tension. While the experiments show their combination is effective, the paper could benefit from a deeper discussion on how these components work synergistically rather than counteracting each other.

**Questions:**

1. Regarding the mask design in TMPD, ehe random mask strategy is simple and effective. Have you explored other, more sophisticated masking approaches, such as learnable masks or masks that are dynamically adapted based on the timestep? Do you think such strategies could offer additional performance gains?
2. Regarding the progressive alignment in TPD, the TPD achieves progressive consistency by aligning the "cumulative average prediction" of consecutive timesteps. How does this compare to a simpler strategy of directly aligning the cumulative prediction at each step with the final prediction (at time T)? Do you think the "step-wise" smoothing is crucial for training stability?
3. Regarding the synergy between TMPD and TPD, as TMPD is designed to introduce "diversity" and TPD is designed to enforce "consistency," two goals that could somehow be in tension. Could you elaborate on a deeper level how these two seemingly opposing goals work synergistically within the MEOM framework?

---

> ### Author Response · Authors · 2025-11-21
>
> ## Response to Weekness 1
> Thank you for the insightful suggestion. We investigated more adaptive and timestep-aware masking strategies, including a learnable Parameter Mask, a Gating Mask, and a Gumbel-Sigmoid Mask. As shown in the table, none of these designs outperform the random mask. We argue that learnable masks tend to follow the student’s intermediate predictions and introduce biased perturbations, whereas the random mask provides simple, unbiased temporal variations that better serve the purpose of TMPD. We have included these results in the discussion section to clarify this design choice and to encourage future exploration of more advanced mask matching strategies for improving teacher–student alignment.
>
> **Table: Comparison of different mask operations on CIFAR-10 / CIFAR-100**
>
> | Mask Operation | Random (Ours) | Parameter | Gating | Gumbel-Sigmoid |
> |----------------|----------------|-----------|--------|-----------------|
> | Acc (%)   | 96.07 / 79.66  | 95.99 / 79.34 | 95.85 / 79.38 | 95.89 / 79.44 |
>
> ## Response to Weekness 2
> Thank you for the insightful question regarding the interplay between TMPD and TPD. As you pointed out, these objectives may appear to be in tension. However, as established by Theorem 2 and Theorem 3 in the main manuscript, the two modules operate on different components of the temporal error decomposition and therefore do not conflict. TMPD improves the quality of the final prediction, while TPD controls how intermediate predictions deviate from this improved final target, enabling the two mechanisms to function cooperatively rather than competitively.
>
> Formally, for any timestep $t<T$, the prediction error decomposes as $\varepsilon(t)=\varepsilon_{\mathrm{full}}+\Delta_{\mathrm{acc}}(t)$. Theorem 2 shows that TMPD strictly reduces the full-length term, $\varepsilon^{\mathrm{TMPD\_full}}<\varepsilon^{\mathrm{TWD\_full}}$, while Theorem 3 shows that TPD strictly reduces the truncation term, in the sense that $\Delta^{\mathrm{TPD\_acc}}(t)<\Delta^{\mathrm{NSC\_acc}}(t)<\Delta^{\mathrm{NC\_acc}}(t)$. Since these two terms are additive and affected independently by the two modules, combining TMPD and TPD yields a strictly smaller total error: $\varepsilon^{\mathrm{MEOM}}(t)=\varepsilon^{\mathrm{TMPD\_full}}+\Delta^{\mathrm{TPD\_acc}}(t)<\varepsilon^{\mathrm{TWD\_full}}+\Delta^{\mathrm{NC\_acc}}(t)=\varepsilon^{\mathrm{baseline}}(t)$. This shows that TMPD and TPD improve complementary parts of the error rather than competing with each other.
> We added the full statement and derivation in the end of Appendix~B.3.
>
> ## Response to Problem 1
> See weakness \#1 response.
>
> ## Response to Problem 2
> To address this question, we added ablations including (1) F-6 A-5, which directly aligns each cumulative prediction with the final prediction at time $T$, and (2) Neighbor-Step Consistency (NSC), which enforces only local alignment between adjacent timesteps. The results show that F-6 A-5 underperforms because the globally averaged signal cannot fully compensate for immature early states, while NSC captures only local consistency. In contrast, TPD performs progressive, step-wise alignment, allowing each timestep to inherit and refine information from the previous one. This gradual propagation smooths the optimization landscape, stabilizes training, and implicitly models longer-range temporal dependencies. We have added these findings to the ablation study.
>
> **Table: Comparison of different temporal step selection strategies in TPD**
>
> | Dataset | R-3 | F-6 R-2 | F-2-4-6 | F-6 A-5 | NSC | **FULL** |
> |---------|-----|---------|---------|---------|------|----------|
> | **C10**   | 96.19 | 96.24 | 96.14 | 95.99 | 96.19 | **96.36** |
> | **C100**  | 80.30 | 80.35 | 80.16 | 80.11 | 80.36 | **80.42** |
>
> ## Response to Problem 3
> See weakness \#2 response.

---

> > ### Author Response · Authors · 2025-11-27
> >
> > Our rebuttal is already more than halfway completed. We would appreciate it if you could share any questions or comments at your convenience. We remain open to further discussion.

---

### Official Review · Reviewer_tdth · 2025-10-30

**Soundness:** 2
**Presentation:** 2
**Contribution:** 2
**Rating:** 2
**Confidence:** 3

**Summary:**

The paper proposes a knowledge distillation method for SNN training, called MEOM (Many Eyes, One Mind). The method consists of two parts: Temporal Multi-Perspective Distillation (TMPD), which creates the teacher's feature for each time-step, and Temporal Progressive Distillation (TPD), which averages activation across different time-steps and calculates CE losses between the average results and targets.
Experiment shows that the method has better accuracy than BKDSNN with Spikformer-8-384 on ImageNet.

**Strengths:**

1. The paper organization is clear.
2. The proposed distillation methods are understandable.
3. The results in CIFAR10/100 is good.

**Weaknesses:**

1. Limited novelty:
The proposed work introduces two techniques to enhance the performance of knowledge distillation. However, the TMPD method only provides a marginal improvement (around 0.3%) on CIFAR-10/100, which is within the range of training variance and thus not convincing. Moreover, the TPD approach is rather straightforward and not inherently tied to the distillation framework. Therefore, it could also be applied to conventional BPTT-based training, which weakens its novelty.

2. Insufficient experimental validation on ImageNet:
The experiments on ImageNet lack sufficient ablation studies, making it difficult to attribute the performance gain solely to the proposed techniques. In addition, the authors should consider evaluating on a larger model, such as the S-8-768 structure, where BKDSNN achieves 79.9% accuracy, to strengthen the evidence.

3. Missing analysis of training overhead:
The proposed distillation-based methods are expected to introduce additional training overhead, especially due to the TMPD methods. The authors should include a comparison of the GPU memory footprint and total GPU hours between the proposed methods and standard distillation baselines.

**Questions:**

Please see the weaknesses.

---

> ### Author Response · Authors · 2025-11-21
>
> ## Response to Weekness 1
> Thank you for your thoughtful comments. We apologize if our earlier description did not clearly convey the contributions of TMPD and TPD.
>
> ### 1. Marginal improvement of TMPD:
> In our original submission, the TMPD results were already obtained by averaging over three independent runs. However, we did not explicitly report the variances, which may have caused confusion. We have now added the corresponding variance information to the manuscript.
> In the meantime, the initially reported improvements were obtained under a relatively large temporal length ($T=6$), where the baseline performance is close to saturation. In such regimes, even small absolute gains are difficult to achieve and often remain meaningful. To address your concern more directly, we additionally conducted experiments under shorter temporal lengths ($T<6$), where the baseline is less saturated. As shown in the table below, the improvements from TMPD become larger and more stable across different time scales.
> | **METHOD**      | **MODEL**   | **T = 2**                   | **T = 4**                   | **T = 6**                   |
> |-----------------|-------------|-----------------------------|-----------------------------|-----------------------------|
> | **MEOM (Ours)** | ResNet-18   | 95.51/77.99 ±0.06/0.08      | 96.07/79.66 ±0.05/0.04      | 96.36/80.42 ±0.10/0.06      |
> |                 | ResNet-19   | 96.65/81.82 ±0.10/0.05      | 97.13/82.85 ±0.07/0.04      | 97.08/83.22 ±0.08/0.14      |
>
> ### 2. Advantage of TPD
> While TPD may appear straightforward at first glance, its contribution lies in how it interacts with TMPD. TPD introduces a progressive temporal self-distillation mechanism that aligns intermediate states toward the final prediction in a sequential manner, which complements TMPD. The key novelty is therefore in the synergistic effect between the two components. This synergy is clearly demonstrated in our ablation study, where removing TPD consistently reduces performance, and further supported by the theoretical analysis in Appendix B.3.
>
> Formally, for any timestep $t<T$, the prediction error decomposes as $\varepsilon(t)=\varepsilon_{\mathrm{full}}+\Delta_{\mathrm{acc}}(t)$. Theorem 2 shows that TMPD strictly reduces the full-length term, $\varepsilon^{\mathrm{TMPD\_full}}<\varepsilon^{\mathrm{TWD\_full}}$, while Theorem 3 shows that TPD strictly reduces the truncation term, in the sense that $\Delta^{\mathrm{TPD\_acc}}(t)<\Delta^{\mathrm{NSC\_acc}}(t)<\Delta^{\mathrm{NC\_acc}}(t)$. Since these two terms are additive and affected independently by the two modules, combining TMPD and TPD yields a strictly smaller total error: $\varepsilon^{\mathrm{MEOM}}(t)=\varepsilon^{\mathrm{TMPD\_full}}+\Delta^{\mathrm{TPD\_acc}}(t)<\varepsilon^{\mathrm{TWD\_full}}+\Delta^{\mathrm{NC\_acc}}(t)=\varepsilon^{\mathrm{baseline}}(t)$. This shows that TMPD and TPD improve complementary parts of the error rather than competing with each other.

---

> > ### Author Response · Authors · 2025-11-21
> >
> > ## Response to Weekness 2
> > Thank you for pointing out the need for more comprehensive validation on ImageNet.
> >
> > ### 1. Ablation studies on Imagenet:
> > In the revised manuscript, we have added ablation studies based on ResNet architectures, as shown in the below table, making it clearer that the observed performance gains can indeed be attributed to our techniques.
> >
> > **Ablation of components on ImageNet under the ResNet-34 SNN backbone (T = 4).**
> >
> > | **TAD** | **TWD** | **TMPD** | **TPD** | **ACC (%)** |
> > |--------|---------|----------|---------|-------------|
> > | ✘ | ✘ | ✘ | ✘ | 66.93 |
> > | ✔ | ✘ | ✘ | ✘ | 68.22 |
> > | ✘ | ✔ | ✘ | ✘ | 71.04 |
> > | ✘ | ✔ | ✔ | ✘ | 71.55 |
> > | ✘ | ✔ | ✘ | ✔ | 71.34 |
> > | ✘ | ✔ | ✔ | ✔ | **71.64** |
> >
> > ### 2. Scaling to larger models:
> > To further strengthen the evidence on larger-scale models, we have also started training the S-8-768 backbone. This model requires significantly longer training time on ImageNet, and the results are still in progress, we agree that scaling to such a strong architecture will further reinforce the empirical conclusions.
> >
> > In addition, aside from BKDSNN, most existing SNNKD studies [1–5] predominantly rely on ResNet-style networks. Introducing a Transformer backbone allows us to test the method under a different architectural paradigm. Across both ResNet and Transformer models, our approach consistently improves performance, indicating that the current set of experiments already provides substantial evidence of the method’s effectiveness.
> >
> > [1] Xu, Qi, et al. "Reversing structural pattern learning with biologically inspired knowledge distillation for spiking neural networks." MM. 2024.
> >
> > [2] Zhao, Xiaochen, et al. "Enhanced Self-Distillation Framework for Efficient Spiking Neural Network Training." NeurIPS. 2025.
> >
> > [3] Yu, Chengting, et al. "Efficient Logit-based Knowledge Distillation of Deep Spiking Neural Networks for Full-Range Timestep Deployment." ICML. 2025.
> >
> > [4] Yu, Kairong, et al. "Temporal separation with entropy regularization for knowledge distillation in spiking neural networks." CVPR. 2025.
> >
> > [5] He, Xiang, et al. "An efficient knowledge transfer strategy for spiking neural networks from static to event domain." AAAI. 2024.

---

> > > ### Author Response · Authors · 2025-11-21
> > >
> > > ## Response to Weekness 3
> > > Thank you for highlighting the need to analyze training overhead. In the original draft, we reported the per-batch training time comparison across different methods, and we have now additionally included GPU memory usage and hours to provide a more complete view of the overhead introduced by our distillation components. We  recognize the additional training overhead introduced by our temporal distillation design. The increase in training time and GPU hours mainly comes from TMPD, as it introduces temporally masked teacher logits and multiple KL-divergence computations across timesteps, with no increase in GPU memory, and this remains much cheaper than training multiple teachers. The detailed results are shown in Tables 6 and 9 in the manuscript.
> > >
> > > **Table 6: Comparison of energy consumption, computation operations, training time, GPU hours, GPU memory, and accuracy on CIFAR-10 with ResNet-18 (T=6).**
> > >
> > > | **METHOD**    | **FRs (%)** | **ACs (M)** | **MACs (M)** | **ENERGY (μJ)** | **TT (ms)** | **GPU h (h)** | **GPU Mem (GB)** | **ACC (%)** |
> > > |---------------|-------------|-------------|--------------|------------------|-------------|----------------|-------------------|-------------|
> > > | ANN           | --          | 0.557       | 549.1        | 1757.2           | 31          | 1.07           | 1.99              | 97.06       |
> > > | VanillaSNN    | 14.21       | 72.500      | 3.342        | 17.946           | 123         | 4.25           | 11.50             | 95.74       |
> > > | KDSNN         | 14.90       | 78.641      | 3.342        | 18.560           | 129         | 4.45           | 11.88             | 95.85       |
> > > | TWSNN         | 15.66       | 83.915      | 3.342        | 19.087           | 130         | 4.49           | 11.88             | 96.16       |
> > > | **Ours**      | **15.98**   | **85.644**  | **3.342**    | **19.260**       | **163**     | **5.63**       | **11.88**         | **96.28**   |

---

> > > > ### Author Response · Authors · 2025-11-27
> > > >
> > > > Our rebuttal is already more than halfway completed. We would appreciate it if you could share any questions or comments at your convenience. We remain open to further discussion.

---

### Official Review · Reviewer_BP7W · 2025-11-02

**Soundness:** 3
**Presentation:** 3
**Contribution:** 3
**Rating:** 6
**Confidence:** 4

**Summary:**

This paper introduces MEOM (Many Eyes, One Mind), a unified knowledge distillation framework for spiking neural networks (SNNs). It integrates two complementary modules: Temporal Multi-Perspective Distillation (TMPD), which enhances temporal diversity via masked teacher features, and Temporal Progressive Distillation (TPD), which gradually aligns truncated and full-length predictions to improve temporal consistency.

**Strengths:**

1.The paper clearly identifies the weaknesses of prior temporal distillation approaches and motivates the need for temporal diversity and consistency.
2.The unification of TMPD (“Many Eyes”) and TPD (“One Mind”) provides an intuitive and theoretically supported structure.
3.Experiments cover multiple benchmarks, showing both performance improvement and robustness under truncated inference.
4.The paper’s exposition and proofs (information gain, convergence robustness) provide reasonable theoretical support for the design choices.

**Weaknesses:**

1. TMPD uses random, static masks. While effective, the design choice appears heuristic. Could adaptive or learnable masks yield further benefit? Consider comparing random vs. learned mask distributions or adding mask diversity analysis.
2. TPD enforces progressive step-wise alignment but may neglect long-range temporal dependencies. A long-range consistency variant or additional analysis on global alignment would make the argument more complete.
3. Assuming the final timestep is globally optimal may not hold universally. A calibration or stability analysis across timesteps could validate this assumption, or adaptive target-timestep selection could be explored.
4. Evaluation focuses on ResNet-style SNN backbones, generalization to more complex or recent SNN backbones (e.g., Spiking Transformer, hybrid architectures) remains unexplored.

**Questions:**

see weaknesses

---

> ### Author Response · Authors · 2025-11-21
>
> ## Response to Weekness 1
> Thank you for the suggestion. We investigated adaptive alternatives to the random static mask, including a learnable parameter mask, a gating mask, and a gumbel-sigmoid mask. As shown in the table, none of these designs outperform the random mask. We argue that learnable masks tend to follow the student’s intermediate predictions and introduce biased perturbations, whereas the random mask provides simple, unbiased temporal variations that better serve the purpose of TMPD. We have included these results in the discussion section to clarify this design choice and to encourage future exploration of more advanced mask matching strategies for improving teacher–student alignment.
>
> **Table: Comparison of different mask operations on CIFAR-10 / CIFAR-100**
>
> | Mask Operation | Random (Ours) | Parameter | Gating | Gumbel-Sigmoid |
> |----------------|----------------|-----------|--------|-----------------|
> | Acc (%)   | 96.07 / 79.66  | 95.99 / 79.34 | 95.85 / 79.38 | 95.89 / 79.44 |
>
> ## Response to Weekness 2
> Thank you for raising this important point regarding long-range temporal dependencies. To address this concern, we conducted additional analyses including (1) F-6 A-5, which directly aligns each cumulative prediction with the final prediction at time $T$, and (2) Neighbor-Step Consistency (NSC), which enforces alignment only between adjacent timesteps. The ablation are shown in the table.
>
> Although TPD aligns timesteps progressively, this step-wise propagation is explicitly designed to make the outputs across all timesteps converge toward a coherent final prediction. Each timestep inherits a more aligned representation from the previous one, allowing information to accumulate and propagate through the temporal dimension. F-6 A-5 underperforms because the globally averaged signal cannot fully compensate for immature early states, while NSC captures only local consistency. In contrast, TPD’s cumulative alignment yields the strongest global temporal consistency and alignment.
>
> **Table: Comparison of different temporal step selection strategies in TPD**
>
> | Dataset | R-3 | F-6 R-2 | F-2-4-6 | F-6 A-5 | NSC | **FULL** |
> |---------|-----|---------|---------|---------|------|----------|
> | **C10**   | 96.19 | 96.24 | 96.14 | 95.99 | 96.19 | **96.36** |
> | **C100**  | 80.30 | 80.35 | 80.16 | 80.11 | 80.36 | **80.42** |
>
>
> ## Response to Weekness 3
> Thank you very much for this insightful comment. We apologize for any ambiguity in the original writing. In the paper, we treated the final timestep as achieving a “one mind’’ effect because in our experiments with a small number of timesteps ($\leq$ 6), this behavior consistently appears. We would also like to note that the method remains valid even without relying on this assumption. Under supervised training, once any prefix average $\bar q_{1:t}$ reaches the Bayes-optimal predictor $q^\star$, progressive alignment propagates this prediction forward, making the final $1-T$ average Bayes-optimal as a direct consequence of the objective.
>
> This mechanism reflects the “one mind’’ effect: the alignment constraints enforce $CE(\bar q_{1:t},\bar q_{1:t+1})=0$, which yields $\bar q_{1:1}=\cdots=\bar q_{1:T}$. Since one prefix matches $q^\star$, all must match it, giving $\bar q_{1:T}=q^\star$. Thus, the final average is Bayes-optimal not due to any assumption about the last step, but because any global minimizer collapses all prefix averages to the same optimal predictor. Your comment helped us clarify this more precisely in the discussion section.
>
> ## Response to Weekness 4
> Thank you for raising this point, we have already evaluated a Spiking Transformer with S-8-384 architecture on ImageNet in the Table 2 in the manuscript and still observe consistent gains, showing that our progressive alignment mechanism generalizes well to more advanced SNN architectures. While ResNet-style SNN backbones are the predominant and widely accepted evaluation setting in existing SNN distillation studies, we agree that evaluating more recent and complex architectures further strengthens the generalization claims.

---

> > ### Author Response · Authors · 2025-11-27
> >
> > Our rebuttal is already more than halfway completed. We would appreciate it if you could share any questions or comments at your convenience. We remain open to further discussion.

---

### Meta-Review · Area_Chair_3XcD · 2026-01-06

**Summary:**

This paper proposes MEOM, a unified temporal knowledge distillation framework for spiking neural networks that addresses two limitations of prior temporal distillation methods, namely static teacher supervision and degraded performance under truncated inference. By combining Temporal Multi Perspective Distillation and Temporal Progressive Distillation, the method introduces diverse temporal supervision while progressively aligning intermediate predictions with the full length prediction. Reviewers generally agree that the motivation is clear, the framework is well structured, and the empirical results demonstrate consistent improvements across multiple datasets and architectures, including ImageNet and Transformer based SNNs.

**Reviewer Concerns:**

The main concerns raised by reviewers relate to the novelty of the individual components, the interpretation of TMPD as temporal supervision versus regularization, the heuristic nature of the masking strategy, and the additional training overhead introduced by temporal distillation. These concerns were largely addressed in the rebuttal and revision. The authors provided additional ablation studies, variance reporting, temporal variance analysis, hyperparameter sensitivity experiments, and training cost measurements. They also clarified the theoretical role of TMPD as an implicit regularizer and strengthened the explanation of how TMPD and TPD act on complementary components of the temporal error. While some reviewers remain cautious about the conceptual novelty, most technical and empirical concerns were adequately resolved.

**Reviewer Scores:**

Reviewer scores evolved positively during the discussion. Multiple reviewers initially rated the paper below the acceptance threshold but explicitly raised their scores after the rebuttal, citing improved clarity, stronger theoretical justification, and more comprehensive experiments. Two reviewers rated the paper marginally above the acceptance threshold, and at least one reviewer indicated a willingness to raise their score to 6 but was unable to update it due to system issues. Overall, the post rebuttal consensus trends toward acceptance.

---

### Decision · Program_Chairs · 2026-01-26

Accept (Poster)